# Optimal Query Complexities for Dynamic Trace Estimation

**David P. Woodruff**
Carnegie Mellon University
dwoodruf@cs.cmu.edu

**Fred Zhang**
UC Berkeley
z0@berkeley.edu

**Qiuyi (Richard) Zhang**
Google Brain
qiuyiz@google.com

## Abstract

We consider the problem of minimizing the number of matrix-vector queries needed for accurate trace estimation in the dynamic setting where our underlying matrix is changing slowly, such as during an optimization process. Specifically, for any $m$ matrices $\mathbf{A}_1, ..., \mathbf{A}_m$ with consecutive differences bounded in Schatten-1 norm by $\alpha$, we provide a novel binary tree summation procedure that simultaneously estimates all $m$ traces up to $\varepsilon$ error with $\delta$ failure probability with an optimal query complexity of $\widetilde{O}(m\alpha\sqrt{\log(1/\delta)}/\varepsilon + m\log(1/\delta))$, improving the dependence on both $\alpha$ and $\delta$ from Dharangutte and Musco (NeurIPS, 2021). Our procedure works without additional norm bounds on $\boldsymbol{A}_i$ and can be generalized to a bound for the $p$-th Schatten norm for $p \in [1, 2]$, giving a complexity of $\widetilde{O}(m\alpha(\sqrt{\log(1/\delta)}/\varepsilon)^p + m\log(1/\delta))$. By using novel reductions to communication complexity and information-theoretic analyses of Gaussian matrices, we provide matching lower bounds for static and dynamic trace estimation in all relevant parameters, including the failure probability. Our lower bounds (1) give the first tight bounds for Hutchinson's estimator in the matrix-vector product model with Frobenius norm error *even in the static setting*, and (2) are the first unconditional lower bounds for dynamic trace estimation, resolving open questions of prior work.

## 1 Introduction

Implicit matrix trace estimation is ubiquitous in numerical linear algebra and arises naturally in a wide range of applications, see, e.g., [25]. In this problem, we are given an oracle which gives us matrix-vector products $\boldsymbol{A}x_1, \boldsymbol{A}x_2, \cdots, \boldsymbol{A}x_m$ for an unknown $n \times n$ square matrix $\boldsymbol{A}$ and queries $x_1, \ldots, x_m$ of our choice, that may be chosen adaptively. In typical applications, one cannot afford to compute the diagonal entries of $\boldsymbol{A}$ explicitly, due to $\boldsymbol{A}$ being implicitly represented and computational constraints. The goal is to efficiently estimate $\operatorname{Tr} \boldsymbol{A}$ using only matrix-vector products.

In machine learning and data science, applications of trace estimation include training Gaussian Processes [8, 11], triangle counting [1], computing the Estrada Index [10, 9], and studying optimization landscapes of deep neural networks from Hessian matrices [12, 26]. In these applications, it is common that $\boldsymbol{A}$ is represented implicitly due to its large memory footprint. For example, while it is possible to compute Hessian-vector products via Pearlmutter's trick [20], it is prohibitive to compute or store the Hessian matrix $\boldsymbol{H}$, see, e.g., [12].

Moreover, $\boldsymbol{A}$ may be a matrix function $f$ of another matrix $\boldsymbol{B}$ in some applications. Since computing $f(\boldsymbol{B})$ is expensive, it is desirable to apply implicit trace estimation. For example, during the training of Gaussian Processes, the marginal log-likelihood contains a heavy-computation term, i.e., the log of the determinant of the covariance matrix, $\log(\det(\mathbf{K}))$, where $\mathbf{K} \in \mathbb{R}^{n \times n}$ and $n$ is the number of data points. The canonical way of computing $\log(\det(\mathbf{K}))$ is via a Cholesky factorization on $\mathbf{K}$, which takes $O(n^3)$ time. Instead, implicit trace estimation methods provide fast algorithms for

36th Conference on Neural Information Processing Systems (NeurIPS 2022).

approximating $\log(\det(\mathbf{K})) = \sum_{i=1}^{n} \log(\lambda_i) = \text{tr}(\log(\mathbf{K}))$ on large-scale data. Therefore, it is important to understand the fundamental limits of implicit trace estimation as the *query complexity*, i.e., the minimum number of matrix-vector multiplications required to achieve a desired accuracy and success rate.

**Static trace estimation and Hutchinson's method.** On the algorithmic side, Hutchinson's method [14] is a simple and widely used method for trace estimation. Let $\mathbf{Q} = [q_1, \ldots, q_\ell] \in \mathbb{R}^{n \times \ell}$ be $\ell$ vectors with i.i.d. standard Gaussian or Rademacher random variables. Given matrix-vector multiplication access to $\mathbf{A}$, Hutchinson's method estimates $\text{tr}(\mathbf{A})$ by $t = \frac{1}{q} \sum_{i=1}^{q} q_i^T \mathbf{A} q_i = \frac{1}{q} \text{tr}(\mathbf{Q}^T \mathbf{A} \mathbf{Q})$. It is known [2] that the estimator satisfies that for any $\varepsilon, \delta \in (0, 1)$,

$$|t - \text{Tr}\, \mathbf{A}| \le \varepsilon \|\mathbf{A}\|_F, \text{ with probability at least } 1 - \delta, \tag{1}$$

provided the number $\ell$ of queries satisfies $\ell \ge C \log(1/\delta)/\varepsilon^2$ for some fixed constant $C$.

For Hutchinson's method, there is also previous work which showed for queries of the form $x^\top \mathbf{A} x$, $\Omega(1/\varepsilon^2)$ queries are required [21]; however, this does not imply even a lower bound for non-adaptive algorithms that use matrix-vector queries. Though stronger algorithmic results and matching lower bounds are known for the important case of PSD matrices in the non-adaptive setting [18, 16], the optimality of Hutchinson's estimator as an trace estimator for general square matrices in the matrix-vector product model still remains an open problem. Notably, Hutchinson's method chooses the query vectors non-adaptively and it is furthermore unclear whether adaptivity could help.

More generally, there has been a flurry of recent work that gives trace estimators with $o(1/\varepsilon^2)$ query complexity but with a different error guarantee. Specifically, let us consider a Schatten-$p$ norm error guarantee, where the goal is to provide an estimate $t$ such that

$$|t - \text{Tr}\, \mathbf{A}| \le \varepsilon \|\mathbf{A}\|_p, \text{ with probability at least } 1 - \delta, \tag{2}$$

where $\|\mathbf{A}\|_p$ denotes the Schatten-$p$ norm.

For $p = 1$, a previous work [18] proposes a variance-reduced version of Hutchinson's method that uses only $O(1/\varepsilon)$ matrix-vector product queries to achieve a nuclear norm error of $\varepsilon \|\mathbf{A}\|_*$, in contrast to the $O(1/\varepsilon^2)$ queries used when the error is in the Frobenius norm. When the matrix is positive semidefinite (PSD), the nuclear norm error is equivalent to a $(1 + \varepsilon)$ multiplicative approximation to the trace. Their work, along with a subsequent work [16], shows that $\Omega(1/\varepsilon)$ queries are therefore sufficient and necessary to achieve a $(1 + \varepsilon)$ multiplicative trace approximation in this setting. While this line of work mainly focuses on PSD matrices and nuclear norm error, we consider trace estimation on general square matrices with Schatten-$p$ norm error for any $p \in [1, 2]$.

Furthermore, we note that the variance-reduced Hutchinson's method splits the queries between approximating the top $O(1/\varepsilon)$ eigenvalues, i.e., by computing a rank-$O(1/\varepsilon)$ approximation to $\mathbf{A}$, and performing Hutchinsons's method on the remainder. Due to the low rank approximation subroutine, the query complexity's dependence on the failure probability is more concretely $O(\sqrt{\log(1/\delta)}/\varepsilon + \log(1/\delta))$ for additive $\varepsilon \|\mathbf{A}\|_*$ error. The additive $\log(1/\delta)$ rate is shown to be necessary when non-adaptive queries are used, but it is an open problem whether adaptive queries can remove the additive $\log(1/\delta)$ term for trace estimation with Schatten-$p$ norm error [16].

This motivates the natural question:

> *Question 1: Is Hutchinson's method optimal in terms of $\varepsilon$ and $\delta$ for static trace estimation of general square matrices, even when adaptivity is allowed? How do we generalize Hutchinson's method for error in general Schatten-$p$ norms?*

**Dynamic trace estimation.** In various applications the input matrix is not fixed. For example, during model training, we need to estimate the trace of a dynamically changing Hessian matrix with respect to some loss function. One may assume that the change at each step is not very large. Motivated by such a scenario, a recent work by Dharangutte and Musco [6] studies dynamic trace estimation.

Formally, let $p \in [1, 2]$ and $\mathbf{A}_1, \mathbf{A}_2, \cdots, \mathbf{A}_m$ be $n \times n$ matrices in a stream such that (1) $\|\mathbf{A}_i\|_p \le 1$ for all $i > 1$, where $\|\cdot\|_p$ denotes the Schatten-$p$ norm, and (2) $\|\mathbf{A}_{i+1} - \mathbf{A}_i\|_p \le \alpha < 1$ for all $i \le m - 1$. The goal is to output a sequence of estimates $t_1, \cdots, t_m$ such that for each $i \in [m]$,

$$|t_i - \text{Tr}\, \mathbf{A}_i| \le \varepsilon, \text{ with probability at least } 1 - \delta, \tag{3}$$

| Upper Bounds | | | | |
|---|---|---|---|---|
| Prior Work | Query Complexity | Matrix Type | Failure Rate | Algorithm Type |
| [2, 22] | $O(\log(1/\delta)/\varepsilon^2)$ | general square | $\delta$ | non-adaptive, $p=2$ |
| [18] | $O(\sqrt{\log(1/\delta)}/\varepsilon + \log(1/\delta))$ | PSD | $\delta$ | adaptive, $p=1$ |
| [18] | $O(\log(1/\delta)/\varepsilon)$ | PSD | $\delta$ | non-adaptive, $p=1$ |
| [16] | $O(\sqrt{\log(1/\delta)}/\varepsilon + \log(1/\delta))$ | PSD | $\delta$ | non-adaptive, $p=1$ |
| **This work** [1] | $O((\sqrt{\log(1/\delta)}/\varepsilon)^p + \log(1/\delta))$ | PSD | $\delta$ | non-adaptive, general $p$ |
| Lower Bounds (Adaptive) | | | | |
| [18] | $\Omega(1/(b+\varepsilon\log(1/\varepsilon)))$ | general square, bit | constant | adaptive, $p=1$ |
| **This work** [2] | $\Omega\left(\frac{1}{\varepsilon^p(b+\log(1/\varepsilon))} + \frac{\log(1/\delta)}{(b+\log\log(1/\delta))}\right)$ | general square, bit | $\delta$ | adaptive, general $p$ |
| **This work** [3] | $\Omega\left((\sqrt{\log(1/\delta)}/\varepsilon)^p\right)$ | general square, ram | $\delta$ | adaptive, general $p$ |
| Lower Bounds (Non-Adaptive) | | | | |
| [18] | $\Omega(1/\varepsilon)$ | PSD, ram | constant | non-adaptive, $p=1$ |
| [16] | $\Omega(\sqrt{\log(1/\delta)}/\varepsilon + \frac{\log(1/\delta)}{\log\log(1/\delta)})$ | PSD, ram | $\delta$ | non-adaptive, $p=1$ |
| **This work** [4] | $\Omega\left(\log^{p/2}(1/\delta)/(\varepsilon^p(b+\log(1/\varepsilon)))\right)$ | general square, bit | $\delta$ | non-adaptive, general $p$ |
| **This work** [5] | $\Omega\left((\sqrt{\log(1/\delta)}/\varepsilon)^p + \frac{\log(1/\delta)}{\log\log(1/\delta)}\right)$ | general square, ram | $\delta$ | non-adaptive, general $p$ |

Table 1: Upper and lower bounds on the query complexity for static trace estimation. In the bit complexity model, each entry of the query vector is specified by $b$ bits, and the dependence on $b$ is necessary.
[1]: A static upper bound generalizing Hutch++ [18] to Schatten-$p$ norm error (Theorem C.1).
[2]: An adaptive lower bound via communication complexity of the Gap Equality and Approximate Orthogonality problem (Theorem 4.1), which combines Theorem D.2 and Theorem D.3, resolving an open problem that $\log(1/\delta)$ queries are required in the adaptive setting.
[3]: An adaptive lower bound via information-theoretic analysis of Gaussian Wigner matrices (Theorem 4.2), showing optimal dependence on $\log(1/\delta)$.
[4]: A non-adaptive lower bound via communication complexity of Augmented Indexing (Theorem F.1), optimal in all parameters up to the bit complexity term.
[5]: A non-adaptive lower bound combining our Theorem 4.2 and the prior result from [16].

via matrix-vector multiplication query access to the first $i$ matrices $(\boldsymbol{A}_j)_{j=1}^i$. Naïvely, one could estimate each $\operatorname{Tr} \boldsymbol{A}_i$ independently using Hutchinson's method. This, however, does not exploit that the changes are bounded at each step. Alternatively, one can rewrite $\operatorname{Tr} \boldsymbol{A}_i$ as $\operatorname{Tr} \boldsymbol{A}_1 + \sum_{i=2}^i \operatorname{Tr}(\boldsymbol{\Delta}_i)$, where $\boldsymbol{\Delta}_i = \boldsymbol{A}_i - \boldsymbol{A}_{i-1}$, by linearity of the trace, and apply Hutchinson's method on each term. Unfortunately, this scheme suffers from an accumulation of errors over the steps.

The prior work [6] is focused on $p = \{1, 2\}$ and improves upon the naïve ideas above. For $p = 1$, the authors give a method that uses $O\left(m\sqrt{\alpha/\delta}/\varepsilon + \sqrt{1/\delta}/\varepsilon\right)$ queries. For $p = 2$, they provide an algorithm with query complexity $O(m\alpha \log(1/\delta)/\varepsilon^2 + \log(1/\delta)/\varepsilon^2)$ and a *conditional* lower bound showing that this is tight. This leaves open the question:

> *Question 2: Can we design improved algorithms for dynamic trace estimation under a general Schatten norm assumption? Can we prove an unconditionally optimal lower bound?*

## 1.1 Our Results

Our work resolves the proposed questions (nearly) optimally, and we next discuss our main results.

**Static trace estimation:** For Question 1, we prove query complexity lower bounds for implicit trace estimation in both bit complexity and real RAM models of computation, resolving the open problem of establishing unconditional lower bounds for the optimality of Hutchinson's method even in the adaptive setting.

To do so, we provide new reductions from classic communication complexity problems, including GAP-EQUALITY and APPROXIMATE-ORTHOGONALITY, to matrix trace estimation. Our main lower bounds demonstrate that $\log(1/\delta)$ queries are always needed even with adaptivity and for general $p$, there is an additional $1/\varepsilon^p$ dependence. A key idea is a communication protocol simulation

using the product of two matrices rather than the sum, as was used in prior work on PSD lower bounds [18].

**Theorem 1.1** (Informal; see Theorem 4.1). *In the bit complexity model, where each entry of each query vector is specified using $b$ bits,*

$$\Omega\left(\frac{1}{\varepsilon^p(b + \log(1/\varepsilon))} + \frac{\log(1/\delta)}{b + \log\log(1/\delta)}\right)$$

*number of adaptive queries is necessary to achieve $\varepsilon\|\boldsymbol{A}\|_p$ error with probability at least $1 - \delta$.*

When adaptivity is not allowed, we give a stronger lower bound (Theorem F.1) of $\Omega(\log^{p/2}(1/\delta)/\varepsilon^p)$. This matches the guarantee of Hutchinson's non-adaptive estimator up to a constant factor, for which random sign vectors suffice and so one can take $b = O(1)$.

We also provide a query complexity lower bound in the real RAM model for general Schatten $p$-norms with $p \in [1, 2]$ by using Gaussian ensembles and controlling the remaining entropy of the distribution conditioned on prior queries. In the special case of $p = 2$ (i.e., Frobenius norm error guarantee), our bound again matches the classic Hutchinson's method up to a constant factor for $p = 2$, and an additive $\log(1/\delta)$ factor for $p < 2$. Note that in the non-adaptive setting, our lower bound in the RAM model can also be improved for $p < 2$ to include a $\log(1/\delta)/\log\log(1/\delta)$ factor. Therefore, this lower bound emphasizes that our dependence on $\log(1/\delta)$ in the $\varepsilon$-dependent term is tight, even in the adaptive setting.

**Theorem 1.2** (Informal; see Theorem 4.2). *In the real RAM model, where the queries are real-valued, for sufficiently small $\varepsilon$ and any $p \in [1, 2]$, $\Omega\left(\left(\sqrt{\log(1/\delta)}/\varepsilon\right)^p\right)$ number of adaptive queries is necessary to achieve $\varepsilon\|\boldsymbol{A}\|_p$ error with probability at least $1 - \delta$.*

On the algorithmic front, we give a matching upper bound for static trace estimation for general Schatten-$p$ norm error for $p \in [1, 2]$. The argument requires a careful balancing of the $\varepsilon$ and $\delta$ parameters in the low rank approximation of the Hutch++ procedure from [18]. See Theorem C.1 for a full statement.

**Dynamic trace estimation:** To answer Question 2, we first give an improved algorithm for dynamic trace estimation that uses a binary tree-based decomposition to estimate all matrix traces with only a small logarithmic overhead. The algorithm improves upon the previous work [6] and gets an optimal dependence on $0 < \alpha, \delta < 1$, up to logarithmic factors. Specifically, for $p = 1$, the prior work gives a method that uses $O\left(m\sqrt{\alpha/\delta}/\varepsilon\right)$ queries for small $\varepsilon$, while our algorithm gives an improved $O(m\alpha\sqrt{\log(1/\delta)}/\varepsilon)$ bound with a linear dependence on $\alpha$ and square root dependence on $\log(1/\delta)$. For $p = 2$, our algorithm matches the query complexity of $O(m\alpha\log(1/\delta)/\varepsilon^2)$ given by previous work. Furthermore, our algorithm works under a general Schatten $p$-norm assumption for any $p \in [1, 2]$:

**Theorem 1.3** (Informal; see Theorem 3.1 and Theorem C.2). *For any $p \in [1, 2]$, there is a dynamic trace estimation algorithm that achieves error $\varepsilon$ and failure rate $\delta$ at each step. The algorithm uses a total of*

$$\widetilde{O}\left((m\alpha + 1)\left(\sqrt{\log(1/(\alpha\delta))}/\varepsilon\right)^p + m\log(1/(\alpha\delta))\right) \tag{4}$$

*matrix-vector product queries. Furthermore, for $p = 1$, it can be improved to*

$$\widetilde{O}\left((m\alpha + 1)\left(\sqrt{\log(1/(\alpha\delta))}/\varepsilon\right) + m\min(1, \alpha/\varepsilon)\log(1/(\alpha\delta))\right) \tag{5}$$

Furthermore, since our algorithm avoids the variance reduction technique from [6], we may relax the assumptions of dynamic trace estimation and require only the first matrix to have norm $\|\boldsymbol{A}_1\| \leq 1$, instead of asking the entire sequence $\boldsymbol{A}_i$ to be bounded in such a way. While the norm bound on all $\boldsymbol{A}_i$ is crucial for the algorithm in [6] (rerunning the analysis naïvely would give a worse query complexity of $O(m^3\alpha^3/\varepsilon)$), our tree-based algorithm achieves a nearly optimal query complexity even when the norm of $\boldsymbol{A}_i$ grows, and we suffer only a $\log m$ overhead in that case. Moreover, in our experiments, we find that our algorithm significantly outperforms previous algorithms on real and synthetic datasets. See Section 6 for our experimental results.

To complement our algorithms, we give unconditional lower bounds showing that our algorithm is nearly optimal. Our lower bounds rely on a reduction from dynamic trace estimation to static matrix trace estimation from [6] and make use of our new lower bounds in the static setting. In particular, the reduction shows that if for a fixed set of parameters $\varepsilon, \delta, p$, a static trace estimation scheme requires $\Omega(r)$ queries, then $\Omega(m\alpha r)$ queries are necessary for any dynamic algorithm. Combining this observation with our static trace estimation lower bounds, we get:

**Theorem 1.4** (Informal; see Theorem 5.2 and Theorem 5.3). *For any $p = [1, 2)$, our algorithm attains the optimal query complexity, up to bit complexity and logarithmic terms.*

More specifically, we prove lower bounds that match the first term in our upper bound (4) for all $p \in [1, 2]$. For $p = 1$, we give a lower bound (Theorem 5.4) matching the the second term in (5) as well, showing that the $m(\log(1/\delta))$ additive dependence is necessary.

For $p = 2$, the prior work [6] gives a upper bound of $O(m\alpha \log(1/\delta)/\varepsilon^2 + \log(1/\delta)/\varepsilon^2)$. Our lower bounds are unconditional and show that the first term is tight. Moreover, the second term is necessary due to the static lower bound when $m = 1$. This result is not contradicted by the claim of Theorem 5.4. In particular, when $\alpha \geq \varepsilon^2$, Theorem 5.4 is weaker than the $\Omega(m\alpha \log(1/\delta)/\varepsilon^2)$ lower bound; and when $\alpha < \varepsilon^2$, the construction by itself requires $\varepsilon/\alpha$ update steps to change the trace by $\varepsilon$, which leads to a lower bound of $\Omega(m\alpha \log(1/\delta)/\varepsilon)$, again weaker than $\Omega(m\alpha \log(1/\delta)/\varepsilon^2)$.

## 2 Preliminaries

A matrix $\boldsymbol{A} \in \mathbb{R}^{n \times n}$ is symmetric positive semi-definite (PSD) if it is real, symmetric and has non-negative eigenvalues. Hence, $x^\top A x \geq 0$ for all $x \in \mathbb{R}^n$. Let $\mathrm{tr}(\boldsymbol{A}) = \sum_{i=1}^n \boldsymbol{A}_{ii}$ denote the trace of $\boldsymbol{A}$. Let $\|\boldsymbol{A}\|_F = (\sum_{i=1}^n \sum_{j=1}^n \boldsymbol{A}_{ij}^2)^{1/2}$ denote the Frobenius norm and $\|\boldsymbol{A}\|_{op} = \sup_{\|\mathbf{v}\|_2=1} \|\boldsymbol{A}\mathbf{v}\|_2$ denote the operator norm of $\boldsymbol{A}$. We let $\|\boldsymbol{A}\|_p = (\sum_i \sigma_i^p)^{1/p}$ be the Schatten-$p$ norm, where $\sigma_i$ are the singular values of $\boldsymbol{A}$. Two special cases are the Frobenius norm, which equals the Schatten-2 norm ($\|\boldsymbol{A}\|_F = \|\boldsymbol{A}\|_2$) and the nuclear norm, equals the Schatten-1 norm ($\|\boldsymbol{A}\|_\star = \|\boldsymbol{A}\|_1$).

## 3 Algorithm for Dynamic Trace Estimation

We give an algorithm for dynamic trace estimation under a general Schatten-$p$ norm assumption, for $p \in [1, 2]$. For $p = 1$, our algorithm provides an improved guarantee upon the DeltaShift++ procedure from [6]. In a later section we complement the result by showing that it is indeed near-optimal. Specifically, we give an algorithm that achieves the following guarantees:

**Theorem 3.1** (Improved dynamic trace estimation). *Let $\boldsymbol{A}_1, \boldsymbol{A}_2, \cdots, \boldsymbol{A}_m$ be $n \times n$ matrices such that (1) $\|\boldsymbol{A}_i\|_\star \leq 1$ for all $i$, and (2) $\|\boldsymbol{A}_{i+1} - \boldsymbol{A}_i\|_\star \leq \alpha$ for all $i \leq m - 1$. Given matrix-vector multiplication access to the matrices, a failure rate $\delta > 0$ and error bound $\varepsilon$, there is an algorithm that outputs a sequence of estimates $t_1, \cdots, t_m$ such that for each $i \in [m]$,*

$$|t_i - \mathrm{Tr}\,\boldsymbol{A}_i| \leq \varepsilon, \text{ with probability at least } 1 - \delta. \tag{6}$$

*The algorithm uses a total of*

$$O\left((m\alpha + 1)\log^2(1/\alpha)\sqrt{\log(1/(\alpha\delta))}/\varepsilon + m\min(1, \alpha/\varepsilon)\log(1/(\alpha\delta))\right) \tag{7}$$

*matrix-vector multiplication queries to $\boldsymbol{A}_1, \boldsymbol{A}_2, \cdots, \boldsymbol{A}_m$.*

Compared with DeltaShift++ in [6], this guarantee provides an exponential improvement in $\delta$ and a polynomial improvement in $\alpha$ for $p \neq 2$, while maintaining the optimal dependence on $m$ and $\varepsilon$.

### 3.1 Algorithm

We now describe our algorithm. The first idea is to partition the $m$ updates into groups of size $s = \lceil 1/(2\alpha) \rceil$. Each group will be treated independently, and we will use

$$O\left(\log^2(1/\alpha)\sqrt{\log(1/(\alpha\delta))}/\varepsilon + \frac{1}{\alpha}\log(1/(\alpha\delta))\right). \tag{8}$$

queries on each group. This leads to our claimed query complexity, as there are $O(m\alpha)$ groups. Note that if $\alpha < \varepsilon$, since $|\operatorname{Tr}(\boldsymbol{A}_j - \boldsymbol{A}_{j-1})| \le \|\boldsymbol{A}_j - \boldsymbol{A}_{j-1}\|_* \le \alpha$, the trace can change by at most an additive $\alpha$, so we can simply ignore every subsequence of length $\varepsilon/\alpha$. Therefore, we only need to apply our estimators to $m\alpha/\varepsilon$ matrices.

Without loss of generality, consider a group of matrices $\boldsymbol{A}_1, \cdots, \boldsymbol{A}_{1/2\alpha}$. As the first step, we estimate $\operatorname{Tr}(\boldsymbol{A}_j - \boldsymbol{A}_{j-1})$ for each $j \ge 2$ by using the Hutch++ static trace estimator [18] as a black box. Then, for each even integer $j = 2k$ (for an integer $2 \le k \le s/2$), we also estimate $\operatorname{Tr}(\boldsymbol{A}_{2k} - \boldsymbol{A}_{2(k-1)})$ in the same way. More generally, for each integer $j = 2^\ell k$, for $0 \le \ell < \log_2 s$, we use Hutch++ to approximate $\operatorname{Tr}\left(\boldsymbol{A}_{2^\ell k} - \boldsymbol{A}_{2^\ell(k-1)}\right)$. We view this scheme as a binary tree: the bottom level consists of leaves corresponding to the trace difference of neighboring matrices, and nodes at level $\ell$ correspond to the trace difference of matrices that are $2^\ell$ apart in their indices.

To output an estimate of $\operatorname{Tr}\boldsymbol{A}_i$, we will write $i$ in its binary representation and approximate it by $\operatorname{Tr}(\boldsymbol{A}_1)$ plus a sequence of $O(\log(1/\alpha))$ differences, at most one for each level in the binary tree. By setting the success rates and errors bounds at each level carefully, we can achieve the desired error guarantee of Equation (6).

To formalize the construction, we first cite the following guarantee of the Hutch++ algorithm:

**Lemma 3.2** (Hutch++, nuclear norm, Theorem 5 of [18])**.** *The Hutch++ estimator uses*

$$N = O\left(\sqrt{\log(1/\delta')}/\varepsilon' + \log(1/\delta')\right)$$

*matrix-vector multiplication queries such that given any square matrix $\boldsymbol{A}$ and parameters $\varepsilon', \delta'$, with probability at least $1 - \delta'$, the algorithm's output $t$ satisfies*

$$|t - \operatorname{Tr}\boldsymbol{A}| \le \sqrt{\varepsilon'}\left\|\boldsymbol{A} - \boldsymbol{A}_{1/\varepsilon'}\right\|_F \le \varepsilon'\|\boldsymbol{A}\|_*. \tag{9}$$

Let $\texttt{Hutch++}(\boldsymbol{A}, \varepsilon', \delta')$ denote the output of Hutch++ on matrix $\boldsymbol{A}$ with parameters $\varepsilon', \delta'$. It will be invoked with different parameters at different levels of the binary tree construction. A description of the algorithm is given by the pseudocode Algorithm 1, with a helper function Algorithm 2.

For simplicity of analysis, note that since we can add dummy matrices (say, extra copies of $\boldsymbol{A}_1$), we assume that each group has size exactly $s = \lceil 1/(2\alpha) \rceil$ and $s$ is a power of two. This blows up the total number of matrices by at most a constant factor.

---

**Algorithm 1:** Improved Dynamic Trace Estimation

---

**Input :** A sequence of square matrices $(\boldsymbol{A}_i)_{i=0}^m \in \mathbb{R}^{n \times n}$, failure rate $\delta$, error bound $\varepsilon$
**Ouput:** Trace estimate $t_i$ for each matrix

1 Partition the matrices into groups of size $s = \lceil 1/(2\alpha) \rceil$.
2 For every $g \ge 0$ and $i \in \{0, 1, \cdots, s-1\}$, let $\boldsymbol{A}_i^{(g)} = \boldsymbol{A}_{gs+i+1}$ denote the $i$-th matrix in the $g$-th group.
3 **for** *each group $\boldsymbol{A}_0^{(g)}, \cdots, \boldsymbol{A}_{s-1}^{(g)}$ independently* **do**
4 $\quad$ Let $t_0 = \texttt{Hutch++}(\boldsymbol{A}_0^{(g)}, \varepsilon/2, \delta/2))$
5 $\quad$ **for** *each level $\ell$ from 0 to $\log_2 s - 1$* **do**
6 $\quad\quad$ $\texttt{gap} = 2^\ell$
7 $\quad\quad$ **for** *$k$ from 1 to $(s-1)/\texttt{gap}$* **do**
8 $\quad\quad\quad$ Compute $t_{k,\ell} = \texttt{Hutch++}\left(\boldsymbol{A}_{k\cdot\texttt{gap}} - \boldsymbol{A}_{(k-1)\cdot\texttt{gap}}, \varepsilon'(\ell), \delta'\right)$, with
$\quad\quad\quad$ $\varepsilon'(\ell) = \varepsilon/(2^{\ell+1}\alpha\log_2 s)$ and $\delta' = \alpha\delta$.
9 $\quad$ Output $t_{gs+i+1} = t_0 + \text{SUMTREE}(1, i, \log_2 s - 1, t)$ for each $i \in [0, s-1]$.

---

## 3.2 Analysis

The analysis of the algorithm is rather lengthy and is delayed to Appendix C.1. In addition, we give a general analysis of the algorithm under Schatten-$p$ norm assumption and the specific improved bounds for $p = 1$ in Appendix C.2 and show how to relax the bounded norm assumption in Appendix C.3.

---

**Algorithm 2:** SUMTREE: Helper Function for Tracing the Binary Tree

**Input:** Indices $i, j$, level $\ell$, binary tree node values $t$

1  gap $= 2^\ell$
2  **if** $j \leq i$ **then**
3     | **return** $0$.
4  **if** $gap = 1$ **then**
5     | **return** $t_{\ell,i}$.
6  **if** $j - i \geq gap$ **then**
7     | **return** $t_{\ell, \lfloor (j-1)/\mathsf{gap} \rfloor} + \text{SUMTREE}(i + \mathsf{gap}, j, \ell - 1, t)$.
8  **else**
9     | **return** $\text{SUMTREE}(i, j, \ell - 1, \mathsf{gap})$.

---

## 4  Lower Bounds for Adaptive Trace Estimation

In this section, we provide (nearly) optimal lower bounds for trace estimation with adaptive matrix-vector multiplication queries, under general square matrices and Schatten-$p$ norm error.

### 4.1  Adaptive Lower Bound, Bit Complexity

First, we show two separate lower bounds under bit complexity model, both proven via reductions from communication complexity problems. One shows an $\Omega(1/\varepsilon^p)$ lower bound (Theorem D.2) and the other $\Omega(\log(1/\delta))$ (Theorem D.3), up to bit complexity terms. Combined together, they yield:

**Theorem 4.1** (Adaptive query lower bound, bit complexity). *Any algorithm that accesses a square matrix $\boldsymbol{A}$ via matrix-vector multiplication queries requires at least*

$$\Omega \left( \frac{1}{\varepsilon^p(k + \log(1/\varepsilon))} + \frac{\log(1/\delta)}{k + \log\log(1/\delta)} \right)$$

*queries to output an estimate $t$ such that with probability at least $1 - \delta$, $|t - \text{Tr}\,\boldsymbol{A}| \leq \varepsilon\|\boldsymbol{A}\|_p$, for any $p \in [1, 2]$, where the query vectors may be adaptively chosen with entries specified by $k$ bits.*

The proofs of the theorems can be found in Appendix D.1.

### 4.2  Adaptive Lower Bound, RAM

Next, we prove a tight lower bound under the real RAM model (Theorem 4.2). The bounds hold for any Schatten-$p$ norm error. Our proof is via information-theoretic analysis of random Gaussian matrices and is delayed to Appendix D.2.

**Theorem 4.2** (Lower Bound for Any Schatten Norm). *For all $p \in [1, 2]$, $\delta > 0$ and $0 < \varepsilon < (\log(1/\delta))^{1/2 - 1/p}$, any algorithm that takes in any input matrix $\boldsymbol{A}$ and succeeds with probability at least $1 - \delta$ in outputting an estimate $t$ such that $|t - \text{tr}(\boldsymbol{A})| \leq \varepsilon\|\boldsymbol{A}\|_p$ requires*

$$m = \Omega \left( \left( \frac{\sqrt{\log(1/\delta)}}{\varepsilon} \right)^p \right)$$

*matrix-vector multiplication queries.*

## 5  Lower Bounds for Dynamic Trace Estimation

Using the query complexity lower bounds for adaptive trace estimation, we can now prove tight lower bounds for dynamic trace estimation. The recent work of Dharangutte and Musco [6] only provides a *conditional* lower bound, assuming that Hutchinson's scheme is optimal. We remove this assumption and make the lower bound unconditional. We additionally prove a lower bound by constructing an explicit hard instance in the dynamic setting. Our lower bounds hold under a general Shatten norm assumption and nearly matches the guarantee of our algorithm.

## 5.1 Lower Bounds via Static-to-Dynamic Reduction

We first show a lower bound for dynamic trace estimation under a Frobenius norm assumption. This immediately implies that the DeltShift algorithm due to [6] is optimal for $p = 2$.

First, we cite a static-to-dynamic reduction from [6] and its implication. The reduction shows how to solve a static instance using a dynamic trace estimation scheme, and therefore any hardness on the static problem translates to the dynamic setting as well. It holds generally for an error bound in any Schatten norm. For completeness, we give a proof in Appendix E.1.

**Lemma 5.1** (Conditional lower bound for dynamic trace estimation [6]). *Suppose that any algorithm that achieves Equation (2) for static trace estimation must use $\Omega(r)$ matrix-vector product queries. Then any dynamic trace estimation algorithm requires $\Omega(r\alpha m)$ matrix-vector product queries under a general Schatten-$p$ norm assumption, when $\alpha = 1/(m-1)$.*

It follows immediately from this lemma and our adaptive query lower bound (Theorem 4.1):

**Theorem 5.2** (Unconditional lower bound for dynamic trace estimation, bit). *For all $p \in [1, 2]$ and $\varepsilon, \delta \in (0, 1)$, any algorithm for dynamic trace estimation under a Schatten-$p$ norm assumption must use at least*

$$\Omega\left(\alpha m \left(\frac{1}{\varepsilon^p(k + \log(1/\varepsilon))} + \frac{\log(1/\delta)}{k + \log\log(1/\delta)}\right)\right)$$

*matrix-vector multiplication queries, where each entry of the query vectors is specified by $k$ bits.*

Combining the same reduction (Theorem D.6) with our previous real RAM lower bound (Theorem 4.2) in the static setting gives:

**Theorem 5.3** (Unconditional lower bound for dynamic trace estimation, RAM). *For all $p \in [1, 2]$, $\delta > 0$ and $0 < \varepsilon < (\log(1/\delta))^{1/2-1/p}$, any algorithm for dynamic trace estimation under a Schatten-$p$ norm assumption must use at least $\Omega\left(\alpha m \left(\sqrt{\log(1/\delta)}/\varepsilon\right)^p\right)$ matrix-vector multiplication queries.*

## 5.2 Lower Bound via Explicit Hard Instance

Using the hard instance based on GAP-EQUALITY in the static setting (from the proof of Theorem D.3), we give an explicit hardness construction against any dynamic trace estimation scheme. This yields the following lower bound, and its proof is in Appendix E.2.

**Theorem 5.4.** *For all $p \in [1, 2]$ and $\varepsilon, \delta \in (0, 1/4)$, any algorithm for dynamic trace estimation under Schatten-$p$ norm assumption must use at least*

$$\Omega\left(m \min\left(1, \frac{\alpha}{\varepsilon}\right) \frac{\log(1/\delta)}{k + \log\log(1/\delta)}\right)$$

*matrix-vector multiplication queries, where each entry of the query vectors is specified by $k$ bits.*

# 6 Experiments

We experimentally validate our algorithmic results. We compare Algorithm 1, with the following procedures on both synthetic and real datasets. More experimental details are in Appendix G.

- Hutchinson's: Apply the classic Hutchinson's scheme for each $\text{Tr}(A_i)$ independently.
- DiffSum: Approximate $t_1 \approx \text{Tr}(\boldsymbol{A}_1)$ and each neighboring difference $d_i \approx \text{Tr}(\boldsymbol{A}_i) - \text{Tr}(\boldsymbol{A}_{i-1})$ using Hutchinson's independently. Then output $t_i = t_1 + \sum_{j=2}^{i} d_j$.
- DeltaShift: The main algorithm of [6]. The experiments from [6] demonstrate that DeltaShift outperforms DiffSum and other Hutchinson-based schemes on various datasets.

**Synthetic data.** We simulate a dynamic trace estimation instance by first generating a (symmetric) random matrix $\boldsymbol{A}^{n \times n}$ and then adding random perturbations over $T = 100$ time steps. The details and results are found in Appendix G.1.

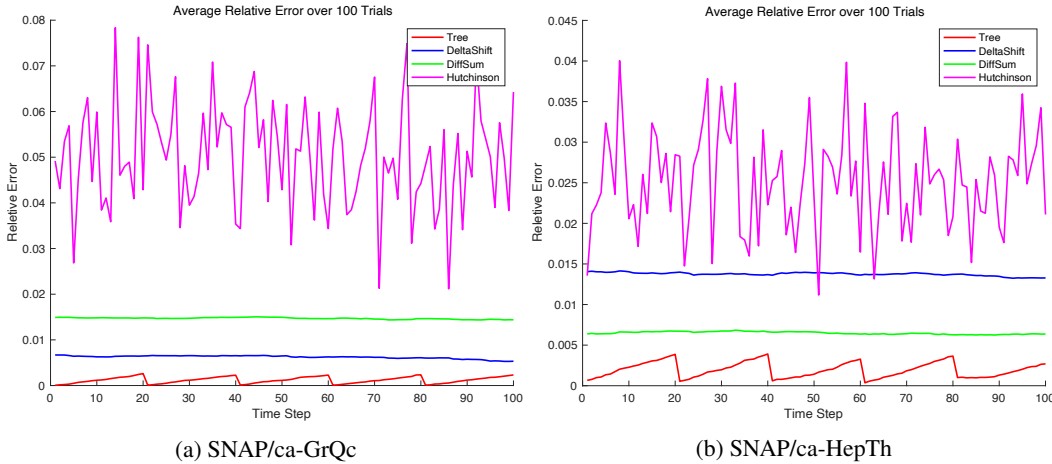

(a) SNAP/ca-GrQc             (b) SNAP/ca-HepTh

Figure 1: ArXiv datasets. Query budget is $8,000$. In this experiment, the trace values are large, so we measure the performance of the algorithms by their relative error $|t_i - \operatorname{Tr} \boldsymbol{A}_i^3| / \max_i \operatorname{Tr} \boldsymbol{A}_i^3$.

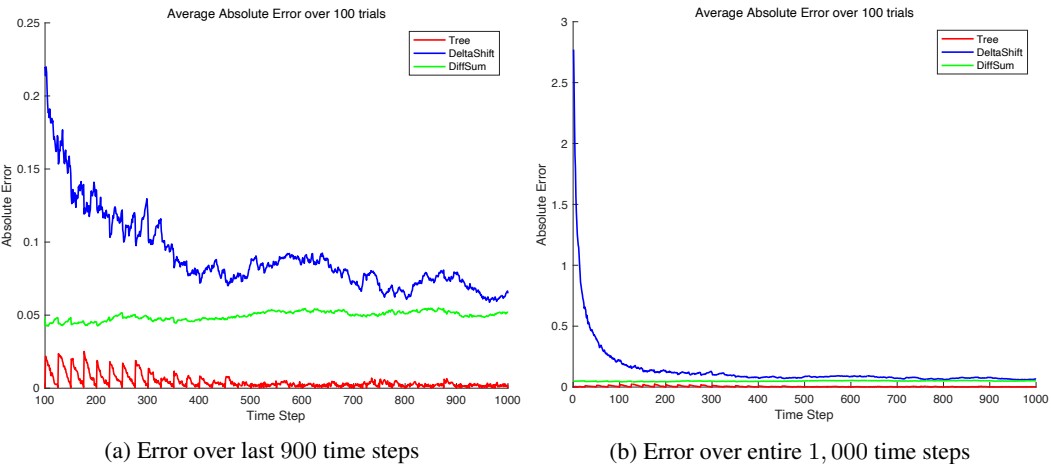

(a) Error over last 900 time steps       (b) Error over entire $1,000$ time steps

Figure 2: MNIST. Query budget is $50,000$.

**Counting triangles.** Our first experiment on a real-world dataset is on counting triangles in dynamic undirected (simple) graphs. Note that the number of triangles in a graph equals $\frac{1}{6} \operatorname{Tr} \boldsymbol{A}^3$, where $\boldsymbol{A}$ is the adjacency matrix of the graph. Thus, triangle counting reduces to trace estimation.

We use two arXiv collaboration networks with $5,242$ and $9,877$ nodes [17].[1] The nodes represent authors, and edges indicate co-authorships. To simulate a real-world scenario, we add a random clique of size at most 6 to the graph in each step, indicating a group of researchers jointly publishing a paper. We note that our algorithm significantly outperforms other methods (Figure 1).

**Neural network weight matrix.** We evaluate the performance of the algorithms on a sequence of weight matrices of a neural network, generated during the training process. In particular, we choose a three-layer neural network with a hidden layer of $100 \times 100$. We train the network on the MNIST dataset via mini-batch SGD and consider the first $1,000$ steps, when the weights are changing most rapidly. Our algorithm achieves much smaller error than DiffSum and DeltaShift (Figure 2).

---

[1]The first is the collaboration network of arXiv General Relativity (ca-GrQc) and the second High Energy Physics Theory (ca-HepTh). Both are available at `https://sparse.tamu.edu/SNAP`.

## Acknowledgement

Work done while David P. Woodruff and Fred Zhang were at Google Research in Pittsburgh.

Fred Zhang is supported by ONR grant N00014-18-1-2562.

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
