# A  More Related Work

We summarize prior work on static trace estimation in Table 1. The seminal work of [2] gives the first analysis of Huchinson's estimator, which was improved by [22]. For PSD matrices, the query complexity can be sharpened, and this was shown recently in [18, 16]. These two papers also give matching lower bounds. The study of dynamic trace estimation was initiated by [6], and our work improves upon their results.

Other applications of implicit trace estimation include inference of Determinantal Point Processes [8], approximating the generalized rank of a matrix [27], computing network centrality measures [3], matrix spectrum estimation [13, 19], and eigenvalue counting [7]. See [25] for a recent survey.

# B  Background on Communication Complexity

Our lower bound proofs use communication complexity. In a communication problem, Alice and Bob receive inputs $x \in \{-1, 1\}^m$ and $y \in \{-1, 1\}^m$, repsectively, and wish to compute a function $f : \{-1, 1\}^m \times \{-1, 1\}^m \to \{-1, 1\}$. The players communicate according to a protocol $P$ and end with an agreed-upon value $z$. The sequence of binary messages exchanged by the players is called the transcript of $P$, denoted $P(x, y)$. We say that the protocol computes $f$ with error $\delta$ if $\Pr(z \neq f(x, y)) \leq \delta$. Let $\mathrm{CC}(P)$ be the length (in bits) of the transcript $P(x, y)$. The communication complexity of $f$ is defined to be the minimum communication cost of any protocol with error $\delta$:

$$\mathrm{CC}_\delta(f) = \min\{\mathrm{CC}(P) : P \text{ computes } f \text{ with error } \delta\}. \tag{10}$$

# C  Proof Details of Section 3

## C.1  Proof of Theorem 3.1

To give a proof sketch, we consider a fixed group and a constant $\delta = \Theta(1)$. To bound the query complexity, we observe that within the group, each level of the binary tree incurs roughly the same number of matrix-vector product queries. Moreover, at the bottom level, there are $s$ calls (including the one computing $t_0$) to Hutch++, with $\varepsilon' = \widetilde{O}(\varepsilon/\alpha)$ and $\delta' = O(\alpha)$. By Theorem 3.2, each call uses $\widetilde{O}(\alpha/\varepsilon)$ queries. Hence, each level uses $\widetilde{O}(s\alpha/\varepsilon) = \widetilde{O}(1/\varepsilon)$ queries. Since there are $\log(1/\alpha)$ levels per group and $O(m\alpha)$ groups, this gives a bound of $\widetilde{O}(m\alpha/\varepsilon)$ on the total number of queries, as claimed in Equation (7). A similar argument shows that the scheme achieves the desired error bound $\varepsilon$ and failure probability $\delta$. We formally prove Theorem 3.1:

*Proof of Theorem 3.1.* Fix a group $g$ and an index $i$. We first argue that the output $t_{gs+i+1}$ is an accurate estimate of the trace $\mathrm{Tr}(\boldsymbol{A}_i^{(g)})$, namely, one which satisfies Equation (6). By construction, the SUMTREE algorithm decomposes the $t_{gs+i+1} - t_0$ into at most $\log_2 s$ terms, one at each level $\ell$. Each term is an estimate $t_{k,\ell}$ of $\mathrm{Tr}\left(\boldsymbol{A}_{2^\ell k}^{(g)} - \boldsymbol{A}_{2^\ell(k-1)}^{(g)}\right)$, for some $\ell, k$. By assumption, each increment $\boldsymbol{A}_i^{(g)} - \boldsymbol{A}_{i-1}^{(g)}$ has Schatten-1 norm at most $\alpha$. Hence, by the triangle inequality,

$$\left\|\boldsymbol{A}_{2^\ell k}^{(g)} - \boldsymbol{A}_{2^\ell(k-1)}^{(g)}\right\|_\star = \left\|\sum_{j=2^\ell k+1}^{2^\ell(k-1)} \boldsymbol{A}_j^{(g)} - \boldsymbol{A}_{j-1}^{(g)}\right\|_\star \leq 2^\ell \alpha$$

By the guarantee of the Hutch++ estimator (Theorem 3.2) and the inequality above, we have that for all $\ell, k$

$$\begin{aligned}
\left|t_{k,\ell} - \mathrm{Tr}\left(\boldsymbol{A}_{2^\ell k}^{(g)} - \boldsymbol{A}_{2^\ell(k-1)}^{(g)}\right)\right| &\leq \varepsilon'(\ell)\left\|\boldsymbol{A}_{2^\ell k}^{(g)} - \boldsymbol{A}_{2^\ell(k-1)}^{(g)}\right\|_\star \\
&\leq \varepsilon'(\ell) \cdot 2^\ell \alpha \\
&= \varepsilon/(2\log_2 s), \tag{11}
\end{aligned}$$

with probability at least $1 - \delta'$. Again, by the guarantee of Hutch++, $t_0$ approximates $\mathrm{Tr}\, \boldsymbol{A}_0^{(g)}$ up to an $(\varepsilon/2)\|\boldsymbol{A}_0^{(g)}\|_\star$ additive error. Therefore, conditioned on Equation (11), the total error of the

estimate $t_{gs+i+1}$ for all $g, i$ is bounded by

$$\left| t_{gs+i+1} - \text{Tr}\left( \boldsymbol{A}_i^{(g)} \right) \right| \leq (\varepsilon/2) \left\| \boldsymbol{A}_0^{(g)} \right\|_\star + (\log_2 s) \cdot \varepsilon/(2\log_2 s)$$
$$\leq (\varepsilon/2) \left\| \boldsymbol{A}_0^{(g)} \right\|_\star + \varepsilon/2$$
$$\leq \varepsilon \tag{12}$$

where the first line follows since there are $\log_2 s$ levels and the last line since $\|\boldsymbol{A}_0^{(g)}\|_\star \leq 1$ by assumption. To bound the failure rate, we note that for a fixed $g$ and $i$, Equation (12) holds if Equation (11) holds for all $t_{k,\ell}$ that are accessed in computing $t_{gs+i+1}$ (via the SUMTREE procedure). By construction, there are at most $\log_2 s$ of these terms, where the bound follows from the number of levels of the binary tree. A simple union bound yields the desired guarantee Equation (6).

It remains to prove the bound on the query complexity (Equation (7)). Each group is treated identically, so we consider any fixed group. Within each group (of size $s$) and at each level $\ell$, we make $O(s/2^\ell)$ calls to $\texttt{Hutch++}(\boldsymbol{A}, \varepsilon'(\ell), \delta')$. By Theorem 3.2, this leads to

$$O\left( (s/2^\ell) \cdot \left( \sqrt{\log(1/\delta')}/\varepsilon'(\ell) + \log(1/\delta') \right) \right)$$

queries at level $\ell$. Plugging in the values of $\varepsilon'(\ell), \delta'$, this equals

$$O\left( \log(1/\alpha)\sqrt{\log(1/(\alpha\delta))}/\varepsilon + \frac{1}{2^\ell \alpha} \log(1/(\alpha\delta)) \right).$$

Summing over $\ell \in \{0, 1, \cdots, \log_2 s - 1\}$, where $s = O(1/\alpha)$, we have that within each group, the number of queries is bounded by

$$O\left( \log^2(1/\alpha)\sqrt{\log(1/(\alpha\delta))}/\varepsilon + \frac{1}{\alpha} \log(1/(\alpha\delta)) \right).$$

There is a total of $\max\{1, O(m\alpha)\}$ groups. Also, we consider the case when $\alpha < \varepsilon$, where the algorithm only needs to provide a fresh estimate every $\varepsilon/\alpha$ time steps. Hence, the sequence length is reduced effectively to $m\alpha/\varepsilon$. Therefore, the query complexity of Algorithm 1 is at most

$$O\left( (m\alpha + 1)\log^2(1/\alpha)\sqrt{\log(1/(\alpha\delta))}/\varepsilon + m\min\{1, \alpha/\varepsilon\}\log(1/(\alpha\delta)) \right).$$

This finishes the proof. $\qquad\square$

## C.2 General Schatten-$p$ Norm Analysis

To generalize trace estimation to $\varepsilon\|\boldsymbol{A}\|_p$ error for any $p \in [1, 2]$, we need to revisit the variance reduction technique to achieve $O(1/\varepsilon)$ query complexity for the nuclear norm. The technique rewrites $\boldsymbol{A} = \boldsymbol{B}_k + \Delta_k$, where $\boldsymbol{B}_k$ is a rank-$k$ matrix with a determined trace, and $\|\Delta_k\|_F \leq O(1)\|\boldsymbol{A} - \boldsymbol{A}_k\|_F$, where $\boldsymbol{A}_k$ is the best rank-$k$ approximation to $\boldsymbol{A}$ for some $k$. Then, we can approximate $\text{tr}(\boldsymbol{A})$ by first explicitly calculating $\text{tr}(\boldsymbol{B}_k)$ and then approximating the trace of $\Delta_k$. The two step procedure requires a careful balancing for how queries are spent between the two components to minimize the total estimation error in the Schatten $p$ norm and results in a $O(1/\varepsilon^p)$ query complexity.

**Theorem C.1** (general Schatten-$p$ error analysis of Hutch++). *The Hutch++ estimator of rank $k$ generalizes to a matrix $\boldsymbol{A}$ with any Schatten norm $p$ bound for $p \in [1, 2]$, and satisfies $|\text{tr}(\boldsymbol{A}) - \texttt{Hutch++}(\boldsymbol{A})| \leq \varepsilon\|\boldsymbol{A}\|_p$ with a total matrix-vector query complexity of*

$$O\left( \left( \frac{\sqrt{\log(1/\delta)}}{\varepsilon} \right)^p + \log(1/\delta) \right)$$

*Proof of Theorem C.1.* Let $\boldsymbol{A}_k$ be the best rank-$k$ approximation to $\boldsymbol{A}$. Then the $\texttt{Hutch++}$ estimator allows us to estimate the trace of $\boldsymbol{A}$ by writing $\boldsymbol{A} = \boldsymbol{A}_k + \Delta$, where $\|\Delta\|_F \leq 2\|\boldsymbol{A} - \boldsymbol{A}_k\|_F$.

Then, we can directly calculate $\mathrm{Tr}(\boldsymbol{A}_k)$ and use Hutchinson's method [14] with $\ell$ matrix-vector multiplication queries, which gives a standard additive error guarantee of

$$C\sqrt{\frac{\log(1/\delta)}{\ell}}\|\Delta\|_F,$$

for some fixed constant $C$. Now, we use the fact that if $\sigma_i$ are the singular values of $\boldsymbol{A}$, then by the definition of Schatten norms,

$$\|\Delta\|_F \leq 2\|\boldsymbol{A} - \boldsymbol{A}_k\|_F \leq \sqrt{\sum_{i=k}^{n}\sigma_i^2} \leq \sqrt{\sigma_k^{2-p}\sum_{i=k}^{n}\sigma_i^p}.$$

Note that we have the following inequality: $k\sigma_k^p \leq \|\boldsymbol{A}\|_p^p$. Therefore, rearranging gives $\sigma_k^{2-p} \leq k^{1-2/p}\|\boldsymbol{A}\|_p^{2-p}$. Finally, we conclude that the total error is bounded by

$$\begin{aligned}
\|\mathrm{tr}(\boldsymbol{A}) - \texttt{Hutch++}(\boldsymbol{A})\| &\leq C\sqrt{\frac{\log(1/\delta)}{\ell}}\|\Delta\|_F \\
&\leq C\sqrt{\frac{\log(1/\delta)}{\ell}}\sqrt{\sigma_k^{2-p}\sum_{i=k}^{n}\sigma_i^p} \\
&\leq C\sqrt{\frac{\log(1/\delta)}{\ell}}\sqrt{k^{1-2/p}\|\boldsymbol{A}\|_p^{2-p}\|\boldsymbol{A}\|_p^p} \\
&\leq C\sqrt{\frac{\log(1/\delta)}{\ell k^{2/p-1}}}\|\boldsymbol{A}\|_p.
\end{aligned}$$

Since we want to set $k = \ell$ to minimize the query complexity, it follows that to reduce to error to $\varepsilon$, we need a number of queries equal to:

$$k = \ell = \left(\frac{\sqrt{\log(1/\delta)}}{\varepsilon}\right)^p.$$

Finally, by the same analysis of [18], using $O(k+\log(1/\delta))$ matrix-vector products suffices to obtain a constant-factor rank-$k$ approximation of $\boldsymbol{A}$, and further, we want $l \geq \log(1/\delta)$. This concludes the proof. $\qquad\square$

With this generalized analysis of Hutch++, one can easily extend Theorem 3.1 and obtain:

**Theorem C.2** (general Schatten-$p$ norm analysis of Algorithm 1). *Let $\boldsymbol{A}_1, \boldsymbol{A}_2, \cdots, \boldsymbol{A}_m$ be $n \times n$ be matrices such that (1) $\|\boldsymbol{A}_i\|_p \leq 1$ for all $i$, and (2) $\|\boldsymbol{A}_{i+1} - \boldsymbol{A}_i\|_p \leq \alpha$ for all $i \leq m-1$ and some $p \in [1,2]$. Given matrix-vector multiplication access to the matrices, a failure rate $\delta > 0$, and an error bound $\varepsilon > 0$, there is an algorithm that outputs a sequence of estimates $t_1, \cdots, t_m$ such that for each $i \in [m]$,*

$$|t_i - \mathrm{Tr}\,\boldsymbol{A}_i| \leq \varepsilon, \text{ with probability at least } 1 - \delta. \tag{13}$$

*The algorithm uses a total of*

$$O\left((m\alpha + 1)\log^{1+p}(1/\alpha)\left(\frac{\sqrt{\log(1/(\alpha\delta))}}{\varepsilon}\right)^p + m\log(1/(\alpha\delta))\right). \tag{14}$$

*matrix-vector multiplication queries to $\boldsymbol{A}_1, \boldsymbol{A}_2, \cdots, \boldsymbol{A}_m$.*

The proof is via counting the number of queries differently using the general Hutch++ analysis Theorem C.1.

*Proof of Theorem C.2.* Our error analysis is almost identical to the nuclear norm case. We sketch it here for completeness. Consider any fixed group $g$. By assumption every increment $\boldsymbol{A}_i^{(g)} - \boldsymbol{A}_{i-1}^{(g)}$

has Schatten-$p$ norm at most $\alpha$. Hence, by the triangle inequality,

$$\left\| \boldsymbol{A}_{2^\ell k}^{(g)} - \boldsymbol{A}_{2^\ell(k-1)}^{(g)} \right\|_p = \left\| \sum_{j=2^\ell k+1}^{2^\ell(k-1)} \boldsymbol{A}_j^{(g)} - \boldsymbol{A}_{j-1}^{(g)} \right\|_p \leq 2^\ell \alpha$$

By the general Schatten-$p$ norm analysis of the Hutch++ estimator (Theorem C.1) and the inequality above, we get that for all $\ell, k$:

$$\begin{aligned}
\left| t_{k,\ell} - \operatorname{Tr}\left( \boldsymbol{A}_{2^\ell k}^{(g)} - \boldsymbol{A}_{2^\ell(k-1)}^{(g)} \right) \right| &\leq \varepsilon'(\ell) \left\| \boldsymbol{A}_{2^\ell k}^{(g)} - \boldsymbol{A}_{2^\ell(k-1)}^{(g)} \right\|_p \\
&\leq \varepsilon'(\ell) \cdot 2^\ell \alpha \\
&= \varepsilon/(2\log_2 s),
\end{aligned} \tag{15}$$

with probability at least $1 - \delta'$. Again, by the guarantees of Hutch++, $t_0$ approximates $\operatorname{Tr} \boldsymbol{A}_0^{(g)}$ up to an $(\varepsilon/2)\|\boldsymbol{A}_0^{(g)}\|_p$ additive error. Therefore, conditioned on Equation (15), the total error of the estimate $t_{gs+i+1}$ for all $g, i$ is bounded by

$$\begin{aligned}
\left| t_{gs+i+1} - \operatorname{Tr}\left( \boldsymbol{A}_i^{(g)} \right) \right| &\leq (\varepsilon/2) \left\| \boldsymbol{A}_0^{(g)} \right\|_p + (\log_2 s) \cdot \varepsilon/(2\log_2 s) \\
&\leq (\varepsilon/2) \left\| \boldsymbol{A}_0^{(g)} \right\|_p + \varepsilon/2 \\
&\leq \varepsilon.
\end{aligned} \tag{16}$$

A union bound thus proves the accuracy guarantee (Equation 13).

We now count the query complexity differently using Theorem C.1. As before, within each group (of size $s$) and in each level $\ell$, we make $O(s/2^\ell)$ calls to $\texttt{Hutch++}(\boldsymbol{A}, \varepsilon'(\ell), \delta')$. This leads to a total number of

$$O\left( (s/2^\ell) \cdot \left( \left( \frac{\sqrt{\log(1/\delta')}}{\varepsilon'(\ell)} \right)^p + \log(1/\delta') \right) \right) \tag{17}$$

matrix-vector multiplication queries by Theorem C.1. Substituting $\varepsilon'(\ell) = \varepsilon/2^{\ell+1}\alpha\log_2 s$ and $\delta' = \alpha\delta$, we have

$$O\left( \log^p(1/\alpha) \left( \frac{\sqrt{\log(1/(\alpha\delta))}}{\varepsilon} \right)^p + + \frac{1}{2^\ell \alpha} \log(1/(\alpha\delta)) \right).$$

Note that we have used the fact that $\alpha 2^{l+1} \leq \alpha s \leq 1$ to simplify the expression. Summing over $\ell \in \{0, 1, \cdots, \log_2 s - 1\}$, where $s = O(1/\alpha)$, we obtain that within each group, the number of queries is bounded by

$$O\left( \log^{1+p}(1/\alpha) \left( \frac{\sqrt{\log(1/(\alpha\delta))}}{\varepsilon} \right)^p + \frac{1}{\alpha} \log(1/(\alpha\delta)) \right).$$

Since there are $\max\{1, O(m\alpha)\}$ groups, the total query complexity is at most

$$O\left( (m\alpha + 1) \log^{1+p}(1/\alpha) \left( \frac{\sqrt{\log(1/(\alpha\delta))}}{\varepsilon} \right)^p + m\log(1/(\alpha\delta)) \right).$$

This completes the proof. $\qquad\square$

## C.3 Relaxing Assumptions

Recall that for dynamic trace estimation, we generally require all matrices $\boldsymbol{A}_i$ to have unit-bounded Schatten-$p$ norm. While it is often the case that the initial matrix $\boldsymbol{A}_1$ has controlled norm, in practice it is unrealistic to assume a general bound on the matrix norm upon dynamic updates. Of course, note that due to the bounded difference assumption, we can always use a linear bound $\|\boldsymbol{A}_i\|_p \leq 1 + \alpha i$. However, using this bound naïvely with the analysis of other algorithms, such as Hutchinson's or its variance-reduced version of [6], introduces additional $\operatorname{poly}(m, \alpha)$ terms in the query complexity. Instead, we show that our tree-based procedure without any initial partitioning still attains an optimal dependence on $m$ and $\alpha$ for the nuclear norm.

**Theorem C.3** (general Schatten-$p$ norm analysis of non-partitioned Algorithm 1). *Let $\boldsymbol{A}_1, \boldsymbol{A}_2, \cdots, \boldsymbol{A}_m$ be $n \times n$ matrices such that (1) $\|\boldsymbol{A}_1\|_* \leq 1$ and (2) $\|\boldsymbol{A}_{i+1} - \boldsymbol{A}_i\|_* \leq \alpha$ for all $i \leq m - 1$. Given matrix-vector multiplication access to the matrices, a failure rate $\delta > 0$ and error bound $\varepsilon$, there is an algorithm that outputs a sequence of estimates $t_1, \cdots, t_m$ such that for each $i \in [m]$,*

$$|t_i - \operatorname{Tr} \boldsymbol{A}_i| \leq \varepsilon, \text{ with probability at least } 1 - \delta. \tag{18}$$

*The algorithm uses a total of*

$$O\left( m\alpha \log(m)^2 \left( \frac{\sqrt{\log(m\delta)}}{\varepsilon} \right) + m \log(m\delta) \right). \tag{19}$$

*matrix-vector multiplication queries to $\boldsymbol{A}_1, \boldsymbol{A}_2, \cdots, \boldsymbol{A}_m$.*

The proof follows by grouping all queries into a group of size $m$, implying that there is a $\log(m)$ overhead by using the tree technique. Therefore, the main alteration to Algorithm 1 is to 1) avoid partitioning into $1/\alpha$ subgroups and 2) calling Hutch++ at each level with updated parameters: $\varepsilon'(\ell) = \varepsilon/(2^{\ell+1}\alpha \log_2 m)$ and $\delta' = \delta/m$.

*Proof of Theorem C.3.* Compared with Theorem 3.1, the error and success rate analysis remains unchanged. We only need to count the query complexity differently using Theorem C.1. Note that in this case, there is only one group of size $s = m$. As before, at each level $\ell$, we make $O(s/2^\ell)$ calls to Hutch++($\boldsymbol{A}, \varepsilon'(\ell), \delta'$). This leads to a total number of

$$O\left( (s/2^\ell) \cdot \left( \left( \frac{\sqrt{\log(1/\delta')}}{\varepsilon'(\ell)} \right)^p + \log(1/\delta') \right) \right) \tag{20}$$

matrix-vector multiplication queries by Theorem C.1. Substituting $\varepsilon'(\ell) = \varepsilon/2^{\ell+1}\alpha \log_2 m$ and $\delta' = \delta/m$, we have

$$O\left( m \log(m)\alpha \left( \frac{\sqrt{\log(m\delta)}}{\varepsilon} \right) + \frac{m}{2^\ell} \log(m\delta) \right).$$

Summing over $\ell \in \{0, 1, \cdots, \log_2 m\}$, we obtain that for this large group, the number of queries is bounded by

$$O\left( m \log^2(m) \left( \frac{\sqrt{\log(m\delta)}}{\varepsilon} \right) + m \log(m\delta) \right).$$

This completes the proof. □

# D  Proof Details of Section 4

## D.1  Proof of Theorem 4.1

We prove the two lower bounds separately. Together they imply Theorem 4.1.

### D.1.1  Lower Bound I

Let $\boldsymbol{A} \in \mathbb{R}^{n \times n}$ be a general square matrix. Recall that the goal is to estimate its trace $\operatorname{Tr} \boldsymbol{A}$ up to an additive $\varepsilon\|\boldsymbol{A}\|_p$. We work under the bit complexity model, where the query vectors $q_1, q_2, \cdots q_r \in \mathbb{R}^n$ have entries specified by $k$ bits. To lower bound $r$, the number of queries, we reduce the communication problem of the APPROXIMATE-ORTHOGONALITY to trace estimation.

The APPROXIMATE-ORTHOGONALITY problem is a two-party communication problem defined on inputs in $\{-1, 1\}^m \times \{-1, 1\}^m$ by the Boolean function

$$\operatorname{ORT}_{b,m}(x, y) = \begin{cases} 1, & \text{if } |\langle x, y \rangle| \leq b\sqrt{m} \\ -1, & \text{otherwise.} \end{cases} \tag{21}$$

The problem is known to have $\Omega(m)$ communication complexity, under the uniform distribution. Let

$$\text{tail}(x) = \frac{1}{\sqrt{2\pi}} \int_x^\infty e^{-x^2/2} dx. \tag{22}$$

be the tail probability of the standard normal.

**Lemma D.1** (Communication complexity of ORT, Theorem 4.2 of [5]). *Let $b > 1/5$ be a constant and $\theta = \text{tail}(2.01 \max\{66, b\})$. Then we have $CC_\theta(\text{ORT}_{b,m}) = \Omega(m)$. The lower bound holds even when the inputs are drawn uniformly from $\{-1,1\}^m \times \{-1,1\}^m$.*

We now prove our adaptive trace estimation lower bound for general matrices, by connecting it with the APPROXIMATE-ORTHOGONALITY problem. It implies that the classic Hutchinson's estimator is optimal for constant success probability.

**Theorem D.2** (Adaptive query lower bound, I). *Any algorithm that accesses a square matrix $\boldsymbol{A}$ via matrix-vector multiplication queries requires at least $\Omega\left(\frac{1}{\varepsilon^p(k+\log(1/\varepsilon))}\right)$ queries to output an estimate $t$ such that with probability at least $1 - \delta/2$, $|t - \text{Tr}\, \boldsymbol{A}| \le \varepsilon \|\boldsymbol{A}\|_p$, for $p \in [1,2]$ and $\delta = \text{tail}(2.01 \cdot 66) = \Theta(1)$, where the query vectors may be adaptively chosen and their entries are specified by $k$ bits.*

*Proof of Theorem D.2.* Let $\mathcal{A}$ be a possibly adaptive algorithm for trace estimation using matrix-vector multiplication queries. Suppose it takes at most $r(n)$ queries to solve the problem, on any $n$-by-$n$ square matrix, with success rate at least $1 - \delta = \Omega(1)$. Consider an instance of APPROXIMATE-ORTHOGONALITY with $b = 2$, where $(x, y)$ is drawn uniformly from $\{-1,1\}^m \times \{-1,1\}^m$.

The proof proceeds by reducing the problem of computing $\text{ORT}_{b,m}(x, y)$ to trace estimation via $\mathcal{A}$. Let $n = \frac{\delta^{p/2}}{2^{p/2}\varepsilon^p} = \Theta(1/\varepsilon^p)$ and $m = n^2$. The reduction and its resulting communication protocol are given as follows. First, given $x \in \{-1,1\}^m$, Alice creates a square matrix $\boldsymbol{A}$, where the rows of $\boldsymbol{A}$ correspond to the entries of $x$ in order. Similarly, given $y \in \{-1,1\}^m$, Bob creates a square matrix $\boldsymbol{B}$, where the columns of $\boldsymbol{B}$ correspond to the entries of $y$ in order. Then the protocol repeats the following steps for $r(n)$ rounds.

   (i) In the $i$-th round from $i = 1$, Alice creates the first query $q_i$, according to $\mathcal{A}$, given all previous query values $\{q_j^\top \boldsymbol{AB}\}_{j<i}$. She computes $q_i^\top \boldsymbol{A}$ and sends it to Bob.

   (ii) Bob computes $q_i^\top \boldsymbol{AB}$ and sends it back to Alice.

At the end of the protocol, with probability at least $1 - \delta/2$, Alice and Bob obtain an estimate $t$ such that

$$|t - \text{Tr}(\boldsymbol{AB})| \le \varepsilon \|\boldsymbol{AB}\|_p, \tag{23}$$

by the guarantee of algorithm $\mathcal{A}$. Finally, they output $z = 1$ if $t \le 3\sqrt{m}$ and $z = -1$ otherwise.

We argue that the above protocol computes $\text{ORT}_{b,m}$ with error at most $\delta = \text{tail}(2.01 \cdot 66)$. First, note that by construction of steps (i) and (ii), we have $\text{Tr}(\boldsymbol{AB}) = \langle x, y \rangle$. Therefore, by Equation (23),

$$\Pr_{\mathcal{A}}(|t - \langle x, y \rangle| \le \varepsilon \|\boldsymbol{AB}\|_p) = \Pr_{\mathcal{A}}(|t - \text{Tr}(\boldsymbol{AB})| \le \varepsilon \|\boldsymbol{AB}\|_p) \ge 1 - \delta/2. \tag{24}$$

It now suffices to show that the error term $\varepsilon \|\boldsymbol{AB}\|_p$ is small. Note that since $x, y$ are drawn uniformly at random, it follows that $\mathbb{E}(\boldsymbol{AB})_{i,j}^2 = n$ for all $i, j \in [n]$. By linearity of expectation, $\mathbb{E}\|\boldsymbol{AB}\|_F^2 = n^3$. By Markov's inequality, $\Pr(\|\boldsymbol{AB}\|_F^2 > tn^3) \le 1/t$ for any $t > 0$, and therefore,

$$\Pr(\varepsilon \|\boldsymbol{AB}\|_F > \varepsilon \sqrt{t} n^{3/2}) \le 1/t.$$

Since $\|\boldsymbol{X}\|_p \le n^{1/p - 1/q} \|\boldsymbol{X}\|_q$ for any $n \times n$ matrix $\boldsymbol{X}$, it follows that

$$\Pr(\varepsilon \|\boldsymbol{AB}\|_p > \varepsilon \sqrt{t} n^{1/p - 1/2} \cdot n^{3/2}) = \Pr(\varepsilon \|\boldsymbol{AB}\|_p > \varepsilon \sqrt{t} n^{1/p+1}) \le 1/t.$$

Plugging in the value of $\varepsilon = \frac{1}{n^{1/p}}\sqrt{\frac{\delta}{2}}$ and setting $t = 2/\delta$, we get

$$\Pr_{x,y}(\varepsilon \|\boldsymbol{AB}\|_p > n) \le 1/t = \delta/2 \tag{25}$$

Combining Equation (24) and Equation (25) and using a union bound,

$$\Pr(|t - \langle x, y \rangle| \le \sqrt{m}) \ge 1 - \delta. \tag{26}$$

Therefore, whenever $\langle x, y \rangle \le 2\sqrt{m}$, we have $t \le 3\sqrt{m}$, and so the protocol outputs $z = 1$ correctly. This proves that the protocol solves $\mathrm{ORT}_{b,m}$ with error at most $\delta$ (for $b = 2$).

To complete the proof, we account for the total communication cost of the protocol. For that, we simply note that each message from Alice or Bob is a vector of $n$ dimensions. It suffices to specify each entry with $k + \log(n/\varepsilon)$ bits. Hence, the protocol solves $\mathrm{ORT}_{b,m}$ with communication cost $r(n) \cdot O(n(k + \log(1/\varepsilon)))$. By the communication lower bound Theorem D.1, it is required that

$$r(n) \cdot O(n(k + \log(1/\varepsilon))) \ge m = n^2.$$

Rearranging and using $n = \Theta(1/\varepsilon^p)$, we have $r(n) \ge \Omega\left(\frac{1}{\varepsilon^p(k + \log(1/\varepsilon))}\right)$, as desired. $\square$

### D.1.2 Lower Bound II

We now give a second lower bound that yields the correct dependence on the failure probability $\delta$. The bound holds for any Schatten-$p$ norm error guarantee, so we state it generally. In particular, we show:

**Theorem D.3** (Adaptive query lower bound, II). *Any algorithm that accesses a square matrix $\boldsymbol{A}$ via matrix-vector multiplication queries requires at least $\Omega\left(\frac{\log(1/\delta)}{k + \log\log(1/\delta)}\right)$ queries to output an estimate $t$ such that with probability at least $1 - \delta$, $|t - \mathrm{Tr}\,\boldsymbol{A}| \le 0.1\|\boldsymbol{A}\|_p$, for any $p$ and any $\delta \in (0, 1)$, where the query vectors may be adaptively chosen and their entries are specified by $k$ bits.*

Our proof leverages another communication problem, GAP-EQUALITY. In this problem, Alice holds $x \in \{0, 1\}^n$ and Bob holds $y \in \{0, 1\}^n$, under the promise that either $x = y$ or $\|x - y\|_2^2 = n/2$. They wish to compute

$$\mathrm{EQ}_n(x, y) = \begin{cases} 1, & \text{if } x = y \\ -1, & \text{otherwise.} \end{cases} \tag{27}$$

The problem requires linear communication complexity for any deterministic protocol [4].

**Lemma D.4** (Communication complexity of GAP-EQUALITY [4]). *Any deterministic protocol for computing $EQ_n$ requires $\Omega(n)$ bits of communication.*

We are now ready to prove Theorem D.3.

*Proof of Theorem D.3.* We give a reduction from solving GAP-EQUALITY as a two-party communication problem to trace estimation via adaptive matrix-vector multiplication queries. Let $n = \log(1/\delta)$ and $x, y \in \{0, 1\}^n$ be an instance of GAP-EQUALITY. Let $\boldsymbol{A} = (x - y)(x - y)^\top$, which has rank 1. Under the promise, either (i) $\boldsymbol{A} = \boldsymbol{0}$, the all 0 matrix, or (ii) has Schatten-$p$ norm $n/2$ for any $p$. In case (ii), we have $\mathrm{Tr}\,\boldsymbol{A} = n/2$. Thus, one can compute $\mathrm{EQ}_n(x, y)$, by estimating $\mathrm{Tr}\,\boldsymbol{A}$ up to an additive error of $0.1\|\boldsymbol{A}\|_p$, for any $p$.

We now argue any trace estimation algorithm $\mathcal{A}$ with failure rate $\delta$ and error $\varepsilon$ yields a deterministic protocol for solving $\mathrm{EQ}_n$. First, by a union bound over all possible $x, y$ under the promise, we have that for all $\boldsymbol{A} = (x - y)(x - y)^\top$, the output $t$ of $\mathcal{A}$ given $\boldsymbol{A}$ always satisfies

$$|t - \mathrm{Tr}\,\boldsymbol{A}| \le \varepsilon\|\boldsymbol{A}\|_p. \tag{28}$$

Suppose $\mathcal{A}$ uses $r = o(\log(1/\delta))$ adaptive queries $q_1, q_2, \cdots, q_r$. In case (i) when $\boldsymbol{A} = \boldsymbol{0}$, all query answers it receives are the zero vector. The algorithm must always output 0, to satisfy the trace estimation guarantee (Equation (28)). Thus, in order to always be correct in case (ii), it must be that one of its query answers is not 0. But as soon as its first query answer is not 0, it knows that it is in case (ii). It follows that algorithm $\mathcal{A}$ just keeps receiving the all-0 vector until it either decides to stop querying or receives a non-zero output vector and immediately decides to stop querying. Thus, for these inputs, we can assume the query algorithm is in fact non-adaptive, since we can consider what its query sequence would be in advance if it were to repeatedly receive the 0 vector as an answer. Hence, we can think of $\boldsymbol{Q} = (q_1, q_2, \cdots, q_r)$ as an $r \times n$ matrix with entries specified with

$k$ bits, and we have the property that $\boldsymbol{Q}(x - y) = 0$ if and only if $x = y$. This gives a protocol for GAP-EQUALITY: Alice simply sends $\boldsymbol{Q}x$ to Bob, who checks if $\boldsymbol{Q}x = \boldsymbol{Q}y$. The communication is $r(k + \log n) = r(k + \log \log(1/\delta))$, which must be $\Omega(\log(1/\delta))$ by Theorem D.4, and so we get an $r = \Omega(\log(1/\delta)/(k + \log \log(1/\delta)))$ adaptive lower bound.

$\square$

## D.2 Proof of Theorem 4.2

We start with a standard definition.

**Definition D.1** (Gaussian and Wigner Random Matrices). *We let $\boldsymbol{G} \sim \mathcal{N}(n)$ denote an $n \times n$ random Gaussian matrix with i.i.d. $\mathcal{N}(0, 1)$ entries. We let $\boldsymbol{W} \sim \mathcal{W}(n) = (\boldsymbol{G} + \boldsymbol{G}^T)/2$ denote an $n \times n$ Wigner matrix, where $\boldsymbol{G} \sim \mathcal{N}(n)$.*

**Fact D.5** (Upper and Lower Gaussian Tail Bounds). *Letting $Z \sim \mathcal{N}(0, 1)$ be a univariate Gaussian random variable, for any $t > 0$, $\Pr[|Z| \geq t] = \Theta(t^{-1} \exp(-\frac{t^2}{2}))$.*

Suppose that we draw a matrix $\boldsymbol{G} \in \mathbb{R}^{n \times n}$ from the Gaussian or related Wigner distribution and try to learn the entries of the matrix via matrix-vector queries. Because the Gaussian is rotationally and subspace invariant, after a few queries, the conditional distribution of the remaining matrix is also Gaussian (or Wigner)-distributed, no matter how the queries are chosen. This property allows us to exactly characterize the remaining uncertainty of the trace estimation procedure, especially with respect to the failure probability $\delta$, even after seeing a few query results.

**Lemma D.6.** *(Conditional Distribution [Lemma 3.4 of [24]]) Let $\boldsymbol{G} \sim \mathcal{N}(n)$ be as in Definition D.1 and suppose our matrix is $\boldsymbol{W} = (\boldsymbol{G} + \boldsymbol{G}^\top)/2$. Suppose we have any sequence of vector queries, $\boldsymbol{v}_1, ..., \boldsymbol{v}_T$, along with responses $\boldsymbol{w}_i = \boldsymbol{W}\boldsymbol{v}_i$. Then, conditioned on our observations, there exists a rotation matrix $\boldsymbol{V}$, independent of $\boldsymbol{w}_i$, such that*

$$\boldsymbol{V}\boldsymbol{W}\boldsymbol{V}^\top = \begin{bmatrix} Y_1 & Y_2^\top \\ Y_2 & \widetilde{\boldsymbol{W}} \end{bmatrix}$$

*where $Y_1, Y_2$ are deterministic and $\widetilde{\boldsymbol{W}} = (\widetilde{\boldsymbol{G}} + \widetilde{\boldsymbol{G}}^\top)/2$, where $\widetilde{\boldsymbol{G}} \sim \mathcal{N}(n - T)$.*

*Proof of Theorem 4.2.* By standard minimax arguments, it suffices to construct a hard distribution for any deterministic algorithm. Consider $\boldsymbol{W} \sim \mathcal{W}(n)$ for some $n$ that we will determine later. From concentration of the singular values of large Gaussian matrices [23], with probability at least $1 - \delta/10$, we have $\sigma_{\max}(\boldsymbol{G}) \leq Cn^{1/2}$ for some absolute constant $C$ when $n \geq \log(1/\delta)$. Therefore, we conclude that $\|\boldsymbol{G}\|_p \leq Cn^{1/2 + 1/p}$ for some absolute constant $C$. Therefore, by the triangle inequality, $\|\boldsymbol{W}\|_p$ can be bounded by the same value.

Let $m$ be the number of matrix-vector queries, and assume that $m \leq n/2$. By Theorem D.6, we see that conditioned on the queries, our matrix $\boldsymbol{W}$ can be decomposed into a determined part and a Gaussian submatrix $\widetilde{\boldsymbol{W}} \sim \mathcal{W}(n - m)$. Therefore, our conditional distribution of the trace of $\boldsymbol{W}$ is, up to a deterministic shift, the same as the distribution of $\widetilde{\boldsymbol{W}}$, which is simply a Gaussian with variance at least $n - m \geq n/2$. We can check this since $\text{tr}(\widetilde{\boldsymbol{W}}) = \frac{1}{2}\text{tr}(\widetilde{\boldsymbol{G}}) + \frac{1}{2}\text{tr}(\widetilde{\boldsymbol{G}}^\top) = \text{tr}(\widetilde{\boldsymbol{G}}) = \sum_i \widetilde{\boldsymbol{G}}_{ii}$, where $\boldsymbol{G}_{ii} \sim N(0, 1)$ are independent for $1 \leq i \leq n - m$.

Since our algorithm determines a Gaussian of variance at least $n - m \geq n/2$ up to an additive error of $\varepsilon\|\boldsymbol{A}\|_p$ with probability at least $1 - \delta$, we conclude that if $\varepsilon\|\boldsymbol{A}\|_p \leq \sqrt{\log(1/\delta)n}$, then we have a contradiction from the anti-concentration of Gaussians (see Theorem D.5). Therefore, whenever $\varepsilon\|\boldsymbol{A}\|_p \leq \sqrt{\log(1/\delta)n}$ holds, we can deduce a lower bound on the number of matrix-vector queries: $m \geq n/2$.

Therefore, solving $\varepsilon\|\boldsymbol{A}\|_p \leq \sqrt{\log(1/\delta)n}$ for the largest possible value of $n$ gives:

$$n = \Omega\left(\left(\frac{\sqrt{\log(1/\delta)}}{\varepsilon}\right)^p\right)$$

Note that this holds for any $\delta, \varepsilon > 0$ such that $n \geq \log(1/\delta)$. Therefore, we need to enforce that $\varepsilon < (\log(1/\delta))^{1/2-1/p}$. $\qquad\square$

# E  Proof Details for Section 5

## E.1  Proof of Theorem D.6

*Proof of Theorem D.6.* Let $\alpha = 1/(m-1)$. Given a square matrix $\boldsymbol{A}$ with $\|\boldsymbol{A}\|_p = 1$, construct a sequence of matrices

$$\boldsymbol{A}_1 = 0, \quad \boldsymbol{A}_2 = \alpha \cdot \boldsymbol{A}, \quad \ldots \quad \boldsymbol{A}_{1/\alpha} = (1-\alpha)\boldsymbol{A}, \quad \boldsymbol{A}_m = \boldsymbol{A}. \tag{29}$$

Suppose that we have a dynamic trace estimation algorithm $\mathcal{A}$ running on the sequence $(\boldsymbol{A}_i)$. By construction, each $\boldsymbol{A}_i$ is a scaling of $\boldsymbol{A}$. Suppose that in the end $\mathcal{A}$ outputs an estimate $t_m$ such that $|t_m - \mathrm{Tr}\,\boldsymbol{A}| \leq \varepsilon \|\boldsymbol{A}\|_p$ with probability at least $1 - \delta$, using matrix-vector multiplies with $\boldsymbol{A}$. This solves the static trace estimation problem with a Schatten-$p$ norm error guarantee. By assumption, it must have used $\Omega(r)$ matrix-vector multiplication queries with respect to $\boldsymbol{A}$. Therefore, if $\mathcal{A}$ uses $o(r\alpha m)$ queries, it would immediately violate our assumption, which is a contradiction. $\qquad\square$

## E.2  Proof of Theorem 5.4

*Proof.* Let $x, y \in \{0,1\}^n$ be an instance of GAP-EQUALITY, where $n = \log(1/\delta)$. Recall that GAP-EQUALITY is a promise problem. Under its promise, either $x = y$ or $\|x - y\|_2^2 = n/2$, and the goal is to distinguish the two cases. For any given $x, y$, let $\boldsymbol{B}_{x,y} = \frac{2}{n}(x-y)(x-y)^\top$. Then since $\boldsymbol{B}_{x,y}$ is rank-1, $\|\boldsymbol{B}_{x,y}\|_p = 0$ if $x = y$ or $\|\boldsymbol{B}_{x,y}\|_p = 1$ otherwise.

To obtain the claimed lower bound, we consider two parameter regimes. First, if $\alpha > \varepsilon$, we construct the following hard instance, which is a sequence of $m$ matrices satisfying the Schatten $p$ norm assumption for dynamic trace estimation. Let $\boldsymbol{A}_0 \in \mathbb{R}^{N \times N}$ be an all 0s matrix, with $N = \min\{m, 1/\alpha\} \log(1/\delta)$. Throughout the updates, $\boldsymbol{A}_i$ will remain a block diagonal matrix, which consists of $m$ block matrices along the diagonal and each of dimension $\log(1/\delta) \times \log(1/\delta)$. In particular, for all steps $i = \{1, 2, \cdots, \min\{m, 1/\alpha\} - 1\}$, we set

$$\boldsymbol{A}_i = \begin{bmatrix} \boldsymbol{B}_1 & 0 & \cdots & \cdots & \cdots & \cdots & 0 \\ 0 & \boldsymbol{B}_2 & 0 & \cdots & \cdots & \cdots & 0 \\ \vdots & 0 & \ddots & \cdots & \cdots & \cdots & \vdots \\ \vdots & \vdots & 0 & \boldsymbol{B}_i & 0 & \cdots & \vdots \\ \vdots & \vdots & \vdots & 0 & 0 & \cdots & \vdots \\ \vdots & \vdots & \vdots & \vdots & \vdots & \ddots & \vdots \\ 0 & 0 & \cdots & \cdots & \cdots & \cdots & 0 \end{bmatrix} \tag{30}$$

where $\boldsymbol{B}_i = \alpha \boldsymbol{B}_{x_i, y_i}$ with $x_i, y_i \in \{0,1\}^n$ an independent instance of GAP-EQUALITY. In other words, at each step $i$, we update $\boldsymbol{A}_{i-1}$ by replacing the $i$-th diagonal block (currently being all 0s) with $\boldsymbol{B}_i$. Each update changes the trace by 0 or $\alpha$, by the construction of $\boldsymbol{B}_{x,y}$. If $m \leq 1/\alpha$, this completes the construction, and note that the matrices $\{\boldsymbol{A}_i\}$ all have norm bounded by 1. If $m > 1/\alpha$, we continue the construction by deleting one distinct diagonal block at each step until the matrix is the zero matrix. Then we repeat the same rounds of insertion (according to Equation (30)) and deletion until reaching time step $m$. Observe again that the construction satisfies the Schatten norm assumption for dynamic trace estimation.

We now argue the query complexity as follows:

- In the case of $m \leq 1/\alpha$, each update is either (i) trivially 0 or (ii) increases the trace by $\alpha > \varepsilon$. Hence, any dynamic algorithm for outputting $|t_i - \mathrm{Tr}\,\boldsymbol{A}_i| \leq \varepsilon < \alpha$, with probability at least $1 - \delta$, would distinguish between case (i) and (ii) with probability at least $1 - \delta$. However, this requires $\Omega\left(\frac{\log(1/\delta)}{k + \log\log(1/\delta)}\right)$ matrix-vector multiplication queries by Theorem D.3.

- In the case of $m \leq 1/\alpha$, note that (almost) half of the update steps are insertions. By the same argument, any dynamic algorithm that gives a good estimate in an insertion step $i$ can solve the hard instance of estimating $\operatorname{Tr} \boldsymbol{B}_i$. Hence, we get the same query complexity lower bound.

To summarize, if $\alpha > \varepsilon$, we get a lower bound of $\Omega\left(m \cdot \frac{\log(1/\delta)}{k + \log\log(1/\delta)}\right)$ queries.

Now we move on to the case of $\alpha \leq \varepsilon$. We use the same construction as described by Equation 30, where each $\boldsymbol{A}_i$ consists of multiple updates over $s = \lceil \varepsilon/\alpha \rceil$ steps by setting $\boldsymbol{B}_i = \sum_{j=1}^{s}(1/s) \cdot \boldsymbol{B}_{x_i,y_i}$ with $\boldsymbol{B}_{x_i,y_i}$ an independent instance of GAP-EQUALITY. We repeat the argument earlier and apply the hardness of Theorem D.3 on the sequence of $\boldsymbol{A}_i$. This blows up the sequence length by a factor of $s$, and hence leads to a lower bound of $\Omega\left(m \cdot \frac{\alpha}{\varepsilon}\frac{\log(1/\delta)}{k+\log\log(1/\delta)}\right)$. $\qquad\square$

## F  Lower Bound for Non-Adaptive Trace Estimation

In the case of non-adaptive queries, we give a stronger lower bound than Theorem 4.1 in the bit complexity model. The bound matches Hutchinson's guarantee for general square matrices up to a bit complexity term.

**Theorem F.1** (Non-adaptive query lower bound). *Any algorithm that accesses a square matrix $\boldsymbol{A}$ via non-adaptive matrix-vector multiplication queries requires at least $\Omega\left(\frac{\log^{p/2}(1/\delta)}{\varepsilon^p(k+\log(1/\varepsilon))}\right)$ queries to output an estimate $t$ such that with probability at least $1 - \delta$, $|t - \operatorname{Tr}\boldsymbol{A}| \leq \varepsilon\|\boldsymbol{A}\|_p$, for any $p$ and $\varepsilon, \delta \in (0, 1)$, where each entry of the query vectors is specified by $k$ bits.*

The proof is via a reduction from the Augmented Indexing communication problem with low error [15]. For a sufficiently large universe $\mathcal{U}$ and an element $\perp \notin \mathcal{U}$, the problem $\mathrm{IND}_{n,\mathcal{U}}$ is defined as follows.

- Alice gets $x = (x_1, x_2, \ldots, x_n) \in \mathcal{U}^n$.
- Bob gets $y = (y_1, y_2, \ldots, y_n) \in (\mathcal{U} \cup \{\perp\})^n$ such that for some unique $i$
  (i) $y_i \in \mathcal{U}$,
  (ii) $y_k = x_k$ for all $k < i$,
  (iii) $y_{i+1} = y_{i+2} = \cdots = y_N = \perp$.

Finally, Bob wishes to output whether $x_i = y_i$. The one-way communication complexity of $\mathrm{IND}_{n,\mathcal{U}}$ is known:

**Lemma F.2** (Communication complexity of Augmented Indexing [15]). *Any one-way communication protocol for computing $\mathrm{IND}_{n,\mathcal{U}}$ with error $\delta \leq \frac{1}{4|\mathcal{U}|}$ requires at least $n \log|\mathcal{U}|/2$ bits of communication.*

We now describe how to solve $\mathrm{IND}_{n,\mathcal{U}}$ in one round of communication via a non-adaptive trace estimation protocol.

*Proof of Theorem F.1.* Let $\kappa = 1/4\delta^{p/2}$, $n = (\sqrt{\log(3/\delta)}/\varepsilon)^p$, $m = c/(4\delta^{p/2}\varepsilon^p)$ for $c > 0$ a small enough constant, and $\mathcal{U} = [\kappa]$. In the following, we view $\mathcal{U}$ equivalently as the collection of one-hot encodings, i.e., 1-sparse vectors in $\{0,1\}^\kappa$. Let $x, y$ be an instance of $\mathrm{IND}_{n,\mathcal{U}}$ and $i$ be the special index under the promise. Given Alice's input $x \in \{0,1\}^{1/\varepsilon^2 \times \kappa}$ and $\varepsilon, \delta \in (0, 1/4)$, we construct an $n \times n$ real square matrix $\boldsymbol{A}$, as follows.

- Let $\boldsymbol{B} \in \{0,1\}^{m \times m}$ have all rows but the $i$-th row being the all-zeros vector;
- The $i$-th row of $\boldsymbol{B}$ is the vector $v = x$ (with precisely $c/\varepsilon^p$ non-zero entries).
- Let $\boldsymbol{A} = \frac{1}{n}\boldsymbol{G}\boldsymbol{B}\boldsymbol{G}^\top$, where $\boldsymbol{G} \in \mathbb{R}^{n \times m}$ is a random matrix with i.i.d. standard Gaussian entries.

To solve $\text{IND}_{n,\mathcal{U}}$, it suffices for Bob to recover $v_i$ with probability at least $1 - \delta$. By construction, we immediately have that $\text{Tr}\,\boldsymbol{B} = v_i$ and $\|\boldsymbol{B}\|_F = \sqrt{c}/\varepsilon^{p/2}$. Moreover, by the guarantee of Hutchinson's estimator (see, e.g., Lemma 2 of [18]),

$$|\text{Tr}\,\boldsymbol{A} - \text{Tr}\,\boldsymbol{B}| \leq \varepsilon^{p/2}(\log(1/\delta))^{1/2-p/2}\|\boldsymbol{B}\|_F = \sqrt{c}(\log(1/\delta))^{1/2-p/2} \leq \sqrt{c}$$

with probability at least $1 - \delta/3$. By the Johnson-Lindenstrauss lemma, $\|\boldsymbol{A}\|_F \leq \sqrt{c}$ with probability $1 - \delta/3$. By construction, $\boldsymbol{B}$ has rank one and so $\boldsymbol{A}$ has rank one. It follows that $\|\boldsymbol{A}\|_p = \|\boldsymbol{A}\|_F \leq \sqrt{c}$ for any $p$.

Now suppose that there is a non-adaptive trace estimation protocol that has $\varepsilon$ approximation error and $\delta/3$ failure rate, using $r$ queries $\boldsymbol{Q} = (q_1, q_2, \cdots, q_r)$. To finish the reduction, Alice sends matrix $\boldsymbol{AQ}$ to Bob. Bob can obtain an estimate $t$ such that with probability at least $1 - \delta/3$, $|t - \text{Tr}\,\boldsymbol{A}| \leq \varepsilon\|\boldsymbol{A}\|_p$. Now taking a union bound and applying the triangle inequality, we have that with probability $1 - \delta$,

$$\begin{aligned} |t - v_i| = |t - \text{Tr}\,\boldsymbol{B}| &\leq |t - \text{Tr}\,\boldsymbol{A}| + |\text{Tr}\,\boldsymbol{A} - \text{Tr}\,\boldsymbol{B}| \\ &\leq \sqrt{c} + \sqrt{c}\varepsilon \\ &< 1/2, \end{aligned}$$

for $\varepsilon < 1/4$ and a sufficiently small $c$ (say, $c < 0.01$). Hence, Bob can recover $v_i$ and compute $\text{IND}_{n,\mathcal{U}}$.

On the other hand, by Theorem F.2, there is a communication lower bound of $\Omega((\log(1/\delta)/\varepsilon^2)$ bits for the problem. Each entry of $v\boldsymbol{Q}$ is specified by $O(\log(1/\varepsilon) + k)$ bits, so the total communication of sending $\boldsymbol{AQ}$ is $O(\log(1/\varepsilon) + k) \cdot r$. This leads to a query lower bound of $r \geq \Omega((\log(1/\delta))/(\varepsilon^2(\log(1/\varepsilon) + k)))$, as claimed. $\qquad\square$

## G  Experimental Details

### G.1  Experimental Results on Synthetic Data

We follow a similar experimental set-up as in [6] and consider small and large perturbations. We also report the average absolute error over all time steps and all trials. In the small perturbation regime, our algorithm achieves errors (average error: $0.0104$) that are negligible in comparison with DeltaShift (average error: $1.9804$) and other procedures. In the high perturbation regime, our algorithm (average error: $1.6607$) outperforms Hutchinson's and Diffsum and is comparable with Deltashift (average error: $1.5868$). We notice, across a variety of regimes, that Hutchinson's estimator and Diffsum tend to accumulate estimation error over the dynamic updates, whereas our algorithm and DeltaShift remain stable.

### G.2  Experimental Setup

**Allocation of query budget.** We allocate the same query budget in each time step of DeltaShift and in Hutchinson's estimator. For DiffSum, we allocate $1/5$ of the budget for estimating $\text{Tr}\,\boldsymbol{A}_1$ and an equal number of queries among the remaining steps. To optimize performance, the number of groups in our algorithm is tuned.

**Experiments on synthetic data.** On both small and large perturbation experiments, we choose the dimension to be $n = 1000$. The first matrix $\boldsymbol{A}_1$ in the sequence is a symmetric matrix with random (unit-norm) eigenvectors and eigenvalues drawn uniformly from $[-1, 1]$. In the small perturbation regime, a random rank-1 matrix $\Delta_j = 5e^{-5}rgg^\top$ is added in each time step, where $r$ is a random sign and $g$ is a standard Gaussian in $n$ dimensions. In the large perturbation regime, each update is a random rank-20 positive semidefinite matrix.

**Neural network weight matrices.** The network consists of two hidden layers of the same size, with standard ReLU activations. The mini-batch size is set to $60$ and learning rate is set to $0.01$.

We optimized the performance of our trace estimation algorithm by choosing its number of groups to be $20$.

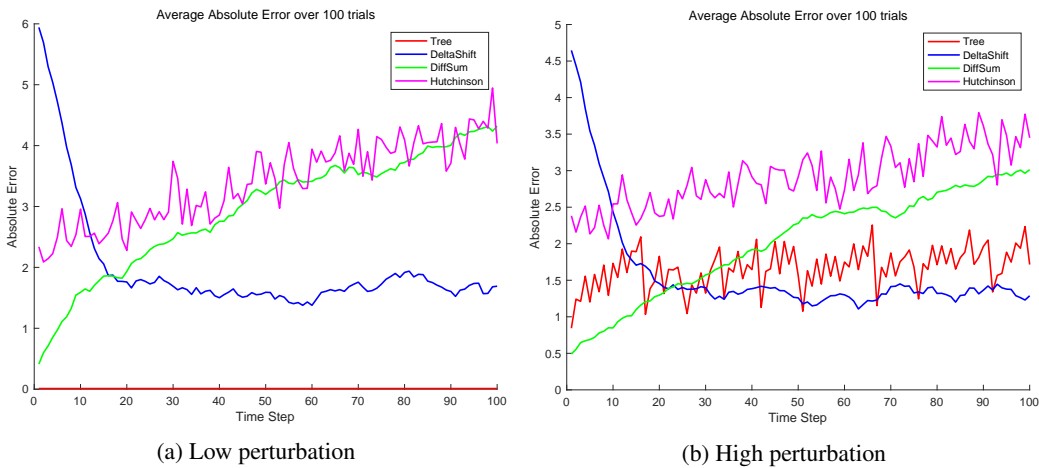

(a) Low perturbation

(b) High perturbation

Figure 3: Synthetic data, Tree refers to our algorithm (Algorithm 1). Query budget is $8,000$. We measure the error at step $i$ simply by absolute error $|t_i - \text{Tr}\, \boldsymbol{A}_i|$