# OpenReview forum: "Optimal Query Complexities for Dynamic Trace Estimation"
_NeurIPS.cc/2022/Conference — NeurIPS 2022 Accept_

### Official Review · Reviewer_5gFw · 2022-07-05

**Rating:** 7
**Confidence:** 4
**Soundness:** 4 excellent
**Presentation:** 3 good
**Contribution:** 3 good

**Summary:**

Estimating the trace of matrices is an important problem in numerical linear algebra. This paper studies the trace estimation problem in a dynamic setting, i.e., we are given m matrices A_1, …, A_m in an online manner, with consecutive differences bounded by alpha in Schatten norms. For bounds under Schatten 1-norm, the authors can improve the dependence on both alpha and delta (failure probability) compared to the previous best-known result in NeurIPS 2021. For the Schattern 2-norm case, the result matches the previous best-known result. As a complementary, the paper also proves matching lower bounds for static and dynamic trace estimation by reductions to communication complexity and information-theoretic analyses of Gaussian matrices.

**Questions:**

It would be helpful if the authors can address my comments above.

**Limitations:**

In the checklist, the authors mark the potential negative societal impact as “not applied”. I in general agree, but at least a brief discussion with 1-2 sentences can be helpful.

**Strengths And Weaknesses:**

From my perspective, the trace estimation problem is of general interest both in theory and in practice. It’s nice to see that the authors can prove tight bounds for the dynamic version of the trace estimation problem. In particular, I think the improvement in the Schatten 1-norm case is a significant contribution – the prior art on this, a NeurIPS paper last year, can only achieve sqrt{alpha} and sqrt{1/delta} complexities, whereas the current paper improves the parameter dependences to alpha and log(1/delta), respectively. This is significant to me. It’s also nice to see that the result can be extended to general Schatten p-norm with p \in [1,2], and also new techniques can be brought to proving matching lower bounds. Moreover, it’s very nice to see very detailed numerical experiments both for synthetic data and real data – many theory papers did not make enough efforts on this to me, but the authors of the current paper made the numerics of their algorithms very persuasive. In all, the technical contributions made by this submission are notable and decent.

In terms of weaknesses, I have the following suggestions:

- It would be helpful to articulate more about the motivation of studying the dynamic trace estimation problem. On the one hand, a main technical contribution of the paper is to study the slow change under Schatten p-norm for p \in [1,2], but why is p \in (1,2) of general interest? Does such general norm appear in real applications? On the other hand, it would also be helpful to describe the real scenarios where we need to estimate the traces of slow-changing matrices.

- In general, I think the authors did a decent job in reviewing prior works, but maybe there’s still space to improve. I have two specific papers to suggest. One is the paper by Sun, Woodruff, Yang, and Zhang in ICALP 2019: https://arxiv.org/pdf/1906.05736.pdf This is an early paper which studies algorithms with matrix-vector products. In particular, Section 3.2 of that paper studied query lower bounds on trace estimation, and it would be clearer if a detailed comparison can be made between that and Section 4 of the current paper (Section 3.1 of this ICALP 2019 paper also talked about adaptivity of matrix-vector queries, which is also relevant).

- The other one is the paper by Braverman, Hazan, Simchowitz, and Woodworth in COLT 2020: http://proceedings.mlr.press/v125/braverman20a/braverman20a.pdf This is also a paper studies matrix-vector queries, and they proved a tight lower bound for linear regression using information-theoretic analyses of Gaussian matrices (see its Section 3). It will be very helpful if the authors can make a comparison between that paper and the idea in the current paper when proving lower bounds.

- Typo: In many places of the paper, the abbreviation dots \cdots should be \ldots. For instance, this is correct in Line 23, but the dots in Line 22 should be \ldots. Such changes can be applied throughout the paper.

---

> ### Author Response · Authors · 2022-07-29
> **Re**
>
> Thank you for the helpful comments! Below we address your questions.
>
> > It would be helpful to articulate more about the motivation of studying the dynamic trace estimation problem. On the one hand, a main technical contribution of the paper is to study the slow change under Schatten p-norm for p \in [1,2], but why is p \in (1,2) of general interest? Does such general norm appear in real applications? On the other hand, it would also be helpful to describe the real scenarios where we need to estimate the traces of slow-changing matrices.
>
>
>   Regarding p in (1,2), consider an n-by-n difference matrix with one singular value equal to n^{1/p} (for some p in (1,2)) and all others 1. Then the 1-norm is O(n), p-norm = O(n^{1/p)}) and 2-norm = O(n^{1/p}). In this case, knowing the choice of p is better than assuming alpha = O(n) and p=1, since we get a smaller alpha. It also betters the assumption that p = 2, since our algorithm’s run-time has an exponent of p (which is shown to be tight up to log factors).
>
> One practical application of dynamic trace estimation is training Gaussian processes, where the kernel matrices may be slowly changing, due to changes in hyperparameters. Each training step requires a log likelihood computation wrt the kernel matrix, which can be approximated by trace estimation method. The introduction section of Dharangutte-Musco also gives more motivating examples.
>
> >  One is the paper by Sun, Woodruff, Yang, and Zhang in ICALP 2019: https://arxiv.org/pdf/1906.05736.pdf This is an early paper which studies algorithms with matrix-vector products. In particular, Section 3.2 of that paper studied query lower bounds on trace estimation, and it would be clearer if a detailed comparison can be made between that and Section 4 of the current paper (Section 3.1 of this ICALP 2019 paper also talked about adaptivity of matrix-vector queries, which is also relevant).
>
> Section 3.2 of the SWYZ proves a lower bound for distinguishing whether a matrix has trace 1 or trace 0. On the other hand, the standard trace estimation problem, considered by our work, asks for an additive approximation up to a factor of eps * || A ||_F. Hence, the lower bound of SWYZ does not translate into our setting. Moreover, their lower bound assumes that the queries are bounded integers, which our RAM lower bounds do not require.
>
> > The other one is the paper by Braverman, Hazan, Simchowitz, and Woodworth in COLT 2020: http://proceedings.mlr.press/v125/braverman20a/braverman20a.pdf
>
> BHSW relies on analyzing the spectrum of the Wishart matrix (W = XX^T, where X has iid Guassian). One of our lower bounds in the RAM model is via an analysis of the Gaussian Wigner matrix, which relies on a similar entropic argument based on partially observed Gaussians. In that sense, ours is more closely related to the lower bound proofs from the prior work by Jiang et al. We will incorporate a comment in a later version.

---

> > ### Comment · Reviewer_5gFw · 2022-08-03
> > **My thoughts after rebuttal**
> >
> > I would like to thank the authors for the detailed replies. I believe that the quality of the paper will be improved after these discussions are adopted. I see that the introduction of the Dharangutte-Musco paper gives motivating examples, but it would be better if the authors can do  a better job on their own introduction (the point on training Gaussian processes is a good one).

---

> > > ### Author Response · Authors · 2022-08-05
> > > **Re:**
> > >
> > > Thank you for the suggestion!  The current submitted version is subject to page limit.  The full paper will contain more    discussions of motivations and related work.

---

### Official Review · Reviewer_2eEg · 2022-07-09

**Rating:** 7
**Confidence:** 4
**Soundness:** 4 excellent
**Presentation:** 2 fair
**Contribution:** 3 good

**Summary:**

The paper looks at query complexity bounds for trace estimation and dynamic trace estimation, in the matrix-vector query model, under Schatten $p$-norm error for $p \in [1, 2]$. In particular, the goal is to obtain an estimate $T$ such that $|T - \text{tr}(A)| \leq \epsilon ||A||_p$. There are four main contributions:

1. Tight lower bounds (in terms of $\epsilon$ and $\delta$) for static trace estimation under Schatten $p$-norm error in two settings: 1. (bounded bit complexity) queries have entries described using $b$ bits and 2. (RAM) query vectors can have real-valued entries. Most notably, the lower bound proves that $\Omega(\log(1/\delta)/\epsilon^2)$ queries are required for general square matrices under Frobenius norm error, even when the queries are adaptively chosen. This proves the tightness of the query complexity of Hutchinson's estimator in terms of both $\epsilon$ and $\delta$. Secondly, in the nuclear norm case (i.e. p = 1), the bound improves on [1] by a $\sqrt{\log(1/\delta)}$ factor.

2. Upper bounds, along with a new algorithm, for dynamic trace estimation under Schatten $p$-norm error. I.e., one is given a sequence of $m$ matrices $A_1, \dots, A_m$ such that $ \text{Schatten p-norm}(A_{i+1} - A_i) \leq \alpha$. The bound is an exponential improvement in the failure probability $\delta$ over [2] in the $p=1$ setting, improving from $\sqrt{1/\delta}$ to $\sqrt{\log 1/\delta}$. Although, there is an additional $\log(m)$ factor. The improvement comes from a new estimator they propose -- by building a binary tree, where the $l$-th level contains estimates of the trace $\text{tr}(A_{2^l(k)} - A_{2^l(k-1)})$.

3. Lower bounds for dynamic trace estimation nearly matching the upper bound are given via a reduction to the static setting.

4. The authors perform experiments on a sparse collaboration network dataset to estimate triangles and on the weight matrix of a NN being trained on MNIST with SGD.

[1] Meyer, R. A., Musco, C., Musco, C., & Woodruff, D. P. (2021). Hutch++: Optimal stochastic trace estimation. In Symposium on Simplicity in Algorithms (SOSA) (pp. 142-155). Society for Industrial and Applied Mathematics.

[2] Dharangutte, P., & Musco, C. (2021). Dynamic Trace Estimation. Advances in Neural Information Processing Systems, 34, 30088-30099.


**Questions:**

- The lower bound in the dynamic setting is claimed to be 'unconditional'. What is it unconditional on? It seems like there are some conditions on the value of $m$ and $\alpha$, i.e., one is fixed in terms of the other.
- In the dynamic setting, does the algorithm require knowledge of $\alpha$? It seems like it does. Is there a simple way to remove this assumption -- a doubling trick maybe? If so, does the algorithm have to re-query vectors in the $i$-th time step with matrices that appeared in previous timesteps?
- It seems like the $log(m)$ factor in the upper bound for the dynamic setting is missing from Theorem 3.1
- What is alpha for the SNAP/MNIST experiments? Can you show some information about how the spectrum of the matrices changes?
- What about how the trace itself changes? How small is the change compared to its value?

**Limitations:**

Yes.

**Strengths And Weaknesses:**

STRENGTHS
The main significance of the results, in my opinion, is a tight lower bound for the query complexity under Frobenius norm error, even allowing for adaptive queries. This proves the optimality of Hutchinson's estimator under Frobenius norm error in the matrix-vector query model.

Since trace estimation is an important primitive in numerical linear algebra/ML and the lower bound (under Frobenius norm error) for general square matrices has remained open, this contribution is a significant result.

On the algorithmic front, in the dynamic setting, the exponential improvement over [2] in the dependence on $\delta$ under nuclear norm error is also an important contribution as it matches the nature of the dependence on $\epsilon$ and $\delta$ in the static setting. The idea required to do this is -- estimating different levels of differences -- is a novel idea over prior work [2].

WEAKNESSES
1. A small weakness of the algorithmic result in the dynamic setting -- it seems like there is an additional $log(m)$ factor in the number of queries, due to the binary tree. This makes the result a little weaker than that of [2] in the Frobenius norm error setting.

2. The author(s) refer to the lower bound in the dynamic setting as 'unconditional' -- although the construction seems like it fixes $\alpha$ in terms of $m$ or vice versa. Please see questions section...

3. Concerns about the writing/clarity: Since there are many results in different regimes -- adaptive vs non-adaptive, bounded bit complexity vs unbounded, static vs dynamic, different values of $p$ -- it is hard to follow each result and how it compares to previous work. Especially since the improvements are in some regimes and not others. I would suggest triaging which results are important in the intro -- Frobenius norm error lower bound in the static setting, dynamic setting upper bound for nuclear norm -- and punting the general theorems to to the appendix. And then clearly state how these are improvements over [1] and [2].

Typos etc.
- footnote 4 in Table 1: 'coplexity' --> complexity
- Algorithm 1 : Line 4, should be two separate lines (?)
- Algorithm 1, 2 : remove the variable 'gap' and just use $2^l$.
- It isn't clear what Algorithm 1 is outputting. Maybe state this at the top?
- Algorithm 1, Input: What is 'i'?
- Theorem 4.1: You started with 'b' for bits and switched to 'k'.
-

---

> ### Author Response · Authors · 2022-07-29
> **Re**
>
>  Thank you for the insightful comments! We will fix the mechanical errors. Detailed response follow.
>
> > The lower bound in the dynamic setting is claimed to be 'unconditional'. What is it unconditional on? It seems like there are some conditions on the value of  m and α, i.e., one is fixed in terms of the other.
>
>  By unconditional, we mean we remove the assumption of Lemma 4.1 of Dharangutte-Musco. In particular, their lower bound assumes that Hutchinson’s estimator cannot be improved for static trace estimation. In this paper, we show that this is true, and therefore makes the lower bound unconditional. We will clarify this point in the final version.
>
> > In the dynamic setting, does the algorithm require knowledge of α? It seems like it does. Is there a simple way to remove this assumption -- a doubling trick maybe? If so, does the algorithm have to re-query vectors in the i-th time step with matrices that appeared in previous timesteps?
>
> In theory, our algorithm only requires an upper bound of $\alpha$. Also since it partitions the updates into chunks of size ~$1/\alpha$ and processes each chunk independently, the choice of $\alpha$ can be set differently for each chunk, as the main use of $\alpha$ is used to simply provide an upper bound on the variance of Hutchinson’s estimator. In practice, we find that it suffices to choose alpha heuristically.
>
> > It seems like the log(m) factor in the upper bound for the dynamic setting is missing from Theorem 3.1
>
>  A log(m) factor would be incurred if we continued the algorithm for all m rounds. Notice that our algorithm partitions the updates into chunks of size ~1/alpha, which leads to a log 1/alpha factor due to the binary tree, rather than log m.
>
>
> > What is alpha for the SNAP/MNIST experiments? Can you show some information about how the spectrum of the matrices changes?
>
> For the SNAP experiment, the alpha is roughly 7 in Frobenius norm (with high probability, since our update matrix is a random clique). A typical matrix in the sequence has norm ~250. In the MNIST experiment, the alpha is roughly 0.013, and a typical matrix in the sequence has norm ~10. In both cases, the matrices are not changing dramatically each step, but still shifting in a non-trivial fashion. Same observation applies to the overall spectrum, and we noticed that our algorithm performs well under such drifts.
>
>
> > What about how the trace itself changes? How small is the change compared to its value?
>
> In most of our experiments, the trace typically changes by 0.05% to 2% each step.

---

> > ### Comment · Reviewer_2eEg · 2022-08-05
> > **Response to author comments**
> >
> > Thank you for all the clarifications. The unconditionality of the lower bound makes sense now.  And I understand that there is no log(m) missing in Thm 3.1.
> >
> > As for the experiments, it might be nice to include the information you included here in the final version.
> >
> > Apart from this, I have no other comments. I'm recommending accept.

---

### Official Review · Reviewer_dsKZ · 2022-07-13

**Rating:** 7
**Confidence:** 3
**Soundness:** 3 good
**Presentation:** 3 good
**Contribution:** 4 excellent

**Summary:**

The paper provides query complexity results for $\epsilon$-trace estimation, with error being multiplicative in some specified matrix norm, under the matrix-vector product oracle in a variety of settings: static/dynamic, adaptive/non-adaptive (the former uses samples that depend on previous ones and therefore, by definition, is sequential; the latter doesn't, and is therefore amenable to parallelization.), error measured in $p$-norm for various $p$, PSD/general square matrices, and combinations of all these. Some specific contributions include

1. improved bounds of the classical Hutch++ algorithm for PSD matrices in general $p$-norm error (previously known only for $p\in \\{1, 2\\}$),

2. lower bounds for adaptive and non-adaptive trace estimation in general $p$-norm error (previously known only for $p\in \\{1, 2\\}$) and, along the way, settling the question of optimality of Hutch++ for $p=2$,

3. (most importantly) a dynamic trace estimation algorithm that improves upon the query complexity for $1$-norm error by Dharangutte-Musco, matches their result for $2$-norm error, and provides a bound (previously unknown) in the range $p \in [1, 2]$.

The lower bounds are proven by reduction to problems in communication complexity, while the dynamic trace estimation improvement is obtained by a binary tree based partition technique combined with the idea of applying Hutch++ to the changes in the input matrices in a specific partition.

**Questions:**

N/A

**Limitations:**

The authors are very clear on what models of computation they use and what regimes their results hold in.

**Strengths And Weaknesses:**

**Strengths**

This is a very strong paper. All the main technical contributions (listed in the previous field) are strong in their own right: one, giving an improved analysis of an existing algorithm, second, showing that this is the best possible query complexity for certain settings of this problem, and third, providing results for previously unknown regimes. Trace estimation --- moreso dynamic trace estimation --- is a task that shows up as a subroutine in several fundamental machine learning algorithms. Therefore, this paper scores highly on significance and originality.

The paper scores highly on clarity: I particularly liked the contextualization of the results in Section 1. I think the algorithm in Section 3.1 could be explained a bit better: in the main body, this currently reads quite terse. (The results on lower bounds could be moved to the Appendix entirely, to fit the page limit, since merely restating the previously mentioned lower bounds formally, but without proofs, doesn't add much value for the reader.)  I also appreciated the clear specification of the computation model (matrix-vector product) along with its justification, the detailed motivation behind the need to study fast trace estimation (through the example of gaussian processes), and the table on related work: to me, this feels like a paper that could easily welcome a new reader into this topic of research.

To summarize, I appreciate the technical contributions and think that the problem is well-motivated. I currently rate this paper 7/10 (GOOD PAPER, ACCEPT).

**Weaknesses**

I think Section 3.1's algorithm could be explained better.

---

> ### Author Response · Authors · 2022-07-27
> **Re**
>
> Thank you for appreciating our work!
>
> We will improve the presentation of section 3.1 in the final version. We acknowledge that there are a few typos, as pointed out by another reviewer. They will be fixed.

---

> > ### Comment · Reviewer_dsKZ · 2022-08-05
> > **Thank you for your response**
> >
> > Thank you for your response, authors! I acknowledge having read your response.
> >
> > Please feel free to let me know if you think I missed any part of your paper (or any significant contribution) that might help me update (increase) your score (either overall or on any of the three specific criteria - soundness, presentation, contribution).

---

### Meta-Review · Area_Chair_ueR2 · 2022-08-26

**Recommendation:** Accept
**Confidence:** Certain

**Metareview:**

The paper provides a novel algorithm with improved complexity bounds for dynamic trace estimation via matrix-vector product queries, assuming bounded differences in Schatten-p norms between consecutive matrices. The paper provides lower bounds proving the optimality of the proposed methods, which are new even in the static setting.

All reviewers appreciated the novelty and technical depth of both the upper and lower bounds  parts of this work, and expect the paper to have significant influence on future research on numerical linear algebra. Consequently, I recommend acceptance, possibly as a spotlight presentation.

**Award:**

No

---

### Decision · Program_Chairs · 2022-09-14

Accept